# When probabilistic seismic hazard *climbs* volcanoes: the Mt Etna case, Italy. Part I: model components for sources parametrization

Raffaele Azzaro[1], Graziella Barberi[1], Salvatore D'Amico[1], Bruno Pace[2], Laura Peruzza[3], Tiziana Tuvè[1]

[1]Istituto Nazionale di Geofisica e Vulcanologia (INGV), Sezione di Catania – Osservatorio Etneo, 95123, Italy
[2] DiSPUTer, Università "G. d'Annunzio" Chieti-Pescara Via dei Vestini, Chieti, Italy
[3]Istituto Nazionale di Oceanografia e di Geofisica Sperimentale – OGS, Sgonico (TS), 34010, Italy

*Correspondence to*: Raffaele Azzaro (raffaele.azzaro@ingv.it)

**Abstract.** The volcanic region of Mt Etna (Sicily, Italy) represents a perfect lab for testing innovative approaches to seismic hazard assessment. This is largely owed to the long record of historical and recent observations of seismic and tectonic phenomena, the high quality of various geophysical monitoring, but also and especially because the rapid geodynamics clearly demonstrate some seismotectonic processes. We present here the model components and the procedures adopted for defining seismic sources to be used in a new generation of Probabilistic Seismic Hazard Assessment (PSHA) whose first results and maps are presented in a companion paper, Peruzza et al. (2017). The sources include, with increasing complexity, seismic zones, individual faults and gridded point sources that are obtained by integrating geological field data with long and short earthquake datasets (the historical macroseismic catalogue that covers about three centuries, and a high-quality instrumental locations database for the last decades). The analysis of the frequency-magnitude distribution identifies two main fault systems within the volcanic complex featuring different seismic rates that are controlled essentially by volcano-tectonic processes. We discuss the variability of the mean occurrence times of major earthquakes along the main Etnean faults by using an historical approach and a purely geologic method. We derive a magnitude-size scaling relationship specifically for this volcanic area, which has been implemented into a recently developed software tool – *FiSH,* Pace et al. (2016) – that we use to calculate the characteristic magnitudes and the related mean recurrence times expected for each fault. Results suggest that for the Mt Etna area, the traditional assumptions of uniform and Poissonian seismicity can be relaxed; a time-dependent fault-based modelling, joined with a 3D imaging of volcano-tectonic sources depicted by the recent instrumental seismicity, can therefore be implemented in PSHA maps. They can be relevant for the retrofitting of the existing building stock, and for driving risk reduction interventions. These analyses do not account for regional M>6 seismogenic sources which dominate the hazard over long return times (≥500 yrs).

## 1 Introduction

Mt Etna, the largest active volcano in Europe, is commonly known for striking volcanic phenomena, featuring nearly constant summit activity and frequent flank eruptions. Less evident but equally impressive are tectonic phenomena occurring along the eastern and southern slopes of the volcano, which are crossed by different systems of active faults (Azzaro et al.,

2012a). The most severe effect of this tectonic activity is the intense seismicity shaking the urbanised areas of the volcano, with obvious implications arising in terms of seismic hazard.

For this reason one of the goals the DPC-INGV V3 Project on the "multi-disciplinary analysis of the relationships between tectonic structures and volcanic activity" (Azzaro and De Rosa, 2016a; 2016b) was to assess seismic hazard in the eastern flank of Etna due to local volcano-tectonic earthquakes. Taking advantage of the huge amount of geological, seismological and geodetic data, which derive from field observations and multi-parametric monitoring, we had the opportunity to test innovative methodological approaches and computation codes developed for the whole of Italy in the framework of previous projects (Azzaro et al., 2012b; Peruzza, 2013), adapting them to consider the features of the volcano-tectonic seismicity. In practice, this meant analysing large seismological and geological datasets to parameterise the seismicity rates of seismic sources and hence the earthquake occurrence probability, thus improving the results of previous researches based solely on the use of macroseismic intensity data (Azzaro et al., 2016, and references therein).

In this study, we present the application of the entire procedure to characterise seismic sources at Mt Etna region providing, for the first time on this volcano, a comprehensive view of the seismotectonic features and an analytical estimation of seismic hazard input parameters, with related uncertainties. The modelling deals with the complexities of the source processes in a volcanic environment, but also with the very nature of the forces controlling the seismicity, which may be by-definition non-Poissonian: we handle them with an increasing degree of detail, in the framework of a logic tree approach. We integrated historical and instrumental earthquake catalogues to define four seismic zones around the main fault systems recognised in the area, identifying the seismogenic layers where most of the seismic energy is released (effective depth) and estimating seismic rates through the frequency-magnitude distribution (FMD). We also used a distributed seismicity model to describe background earthquakes in the crustal volume beneath Mt Etna by adopting a high-resolution three-dimensional grid (inter-nodal distance of 2 km). In addition, we performed the characterisation of the sources at the scale of the individual faults by applying a purely geological approach (Pace et al., 2016) that considers the geometric-kinematic parameters representing fault activity (dimensions and slip-rate). To this end, we first obtained a magnitude-size scaling relationship, specifically for this volcanic region, and then calculated the seismic rates expressed in terms of mean recurrence time of the maximum magnitude expected on each fault.

The obtained dataset defines the input parameters used in a full Probabilistic Seismic Hazard Assessment (PSHA) at a local scale of Mt Etna, discussed in a companion paper (Peruzza et al. 2017, hereinafter referred as Part II). Finally, we remark that the amount of multi-disciplinary data used here for source parametrization, as well as the consistency of results, in our opinion represent a unique condition compared to both other volcanic districts and tectonic areas worldwide. In this sense, we hope that this work may be regarded as a pivot study for improving methodological approaches and conceptual procedures in fault-based, time-dependent hazard estimations.

## 2 Linking faults to earthquakes at Mt Etna: a short overview

Mt Etna volcano is an ideal lab for observing at a small-scale a full range of faulting processes that are difficult to find taking place together in other regions. Evidence of active tectonics is impressive and widespread, particularly in the eastern flank where morpho-tectonic features (Azzaro et al., 2012a), recurrent seismicity (examined extensively in Sections 3 and 4) and ground deformations (Bonforte et al., 2011; Bruno et al., 2012) provide a real measure of the intense volcano-tectonic activity. Furthermore, more than a century of documented history of surface faulting related to coseismic displacements (Azzaro, 1999) and creeping phenomena (Rasà et al., 1996), suggests a clear picture of the relationship between faults and earthquakes; the long list of large and minor events rupturing different segments of the same faults (Fig. 1a) has led to a detailed mapping of active faults and characterisation of their behaviour both in the long- and short-term (Azzaro et al., 2013a).

All these features have allowed constraining a seismotectonic model (Azzaro, 2004) where faults slip in a strongly heterogeneous mode along strike, with two end-member rupture mechanisms addressing fault segments ruled by stick-slip behaviour (earthquake-related slip) or by stable-sliding behaviour (aseismic creeping) (Fig. 1b). In this framework, the Timpe system and the Pernicana fault are the most important tectonic elements at Mt Etna, dissecting the eastern flank between the coast and the volcano-tectonic structures of the NE Rift and Valle del Bove depression (Azzaro et al., 2012a, 2013a and references therein). They are both very active from a seismotectonic point of view, in terms of the number of earthquakes and maximum magnitudes. Whilst the long-term seismic history of the Pernicana fault is limited to a few decades – the urbanisation of the uphill sector crossed by the fault dates back to the late 1970s – the Timpe system is responsible for most of the strongest earthquakes known to have occurred at Mt Etna since the early 19$^{th}$ century: from a total of twelve largest events – here we consider those having epicentral intensities $I_0$ larger than degree VIII EMS, i.e. producing at least severe damage according to the European Macroseismic Scale (see Grünthal, 1998) – ten of them are located here, thus making this densely inhabited zone of Mt Etna the most hazardous of the volcano (Azzaro et al., 2016).

Another debated issue in modelling seismic sources of volcanic regions for seismic hazard applications is the question of whether fault behaviour is strictly speaking controlled by the volcanic activity. It is true that historical destructive earthquakes in the Timpe area did occur both during flank eruptions as well as during periods of volcanic quiescence; correlation studies of major seismic events and volcanic activity have not produced univocal results (Gasperini et al., 1990; Nercessian et al., 1991; Gresta et al., 1994). The inter-event times (IET) statistical analysis (Sicali et al., 2014) shows that the occurrence of low magnitude (M<3) shallow earthquakes in the central sector of the volcano, beneath the summit craters, depends mainly on the volcanic activity and produces a seismicity clustered in space and time. In fact, seismic swarms located here before the 2001, 2002, 2004 and 2008 eruptions, are interpreted as a consequence of stress field variations induced by the process of magma rising and dyke emplacement (Bonaccorso et al., 2004; Gambino et al., 2004; Alparone et al., 2012; Sicali et al., 2015). Conversely, for the flanks of the volcano, especially the eastern one hosting the seismic sources relevant to seismic hazard, the IET distribution shows a prevalence of uncorrelated events, i.e. behaviour more similar to a

tectonic domain than a volcanic one (see Traversa and Grasso, 2010; Bell and Kilburn, 2012). The role of a different, wider stress field acting in the Timpe area - a structurally homogeneous domain characterised by a general east-west extension (Bousquet and Lanzafame, 2004) - is also proved by the analyses of long time series of geodetic and seismic data (Bruno et al., 2012; Solaro et al., 2010; Bozzano et al., 2013; Palano, 2016 and references therein), highlighting the influence of large-scale instability processes where the strain is released by a steady process on decennial time scale (Bonforte et al., 2011). In this scenario we therefore assume that modelled faults are constantly (on average) loaded in time as expected in a typical tectonic process. We also consider the role of eruptive activity controlling the occurrence of low magnitude earthquakes in the central-summit area of the volcano to be negligible for seismic hazard purposes, since this is an uninhabited zone and hence the risk is very low.

### 3 Historical seismicity: some hints for long-term fault behaviour

Information provided from macroseismic data is representative of the long-term seismicity since the effects of past major earthquakes affecting the urbanised areas of the volcano are well documented (Azzaro et al., 2000; Azzaro and Castelli, 2015). The historical earthquake dataset used for the analysis is the CMTE catalogue (CMTE Working Group, 2017), covering the period 1600-2013 and also including fore- and aftershocks of low intensity; overall, nearly 1,800 events are listed in the catalogue. The magnitude of completeness of this macroseismic catalogue has been estimated as $M_c$ 3.7: it corresponds to an epicentral intensity $I_0$ VII EMS (i.e. moderate damage), according to the relationship derived by Azzaro et al. (2011).

For our analysis, we selected the historical earthquakes located along the Timpe fault system (Fig. 2), limiting our attention to the strongest events with $I_0$ ranging from VIII to IX-X EMS (i.e. from severe damage up to destruction) and with a moment magnitude $M_W$ from 4.6 to 5.2. It should be noted that moderate values of magnitude for heavily damaging events are a feature of seismicity in active volcanic areas such as Mt Etna (Azzaro et al., 2011), whereas in tectonic domains crustal earthquakes producing the same effects are generally associated with $M \geq 6$ (Rovida et al., 2016). The main reasons for this behaviour are i) the extremely shallow focal depths of Etna earthquakes (0-4 km, see Section 4.1.1) compared with those of regional events (typically in the range 10-15 km), ii) an anomalously strong low-frequency ($0.1 < f < 1$ Hz) radiation deviating from the conventional Brune (1970) spectral scaling, that causes large ground displacements and long ($\approx$20 sec) durations of shaking (Milana et al., 2008). The final dataset therefore covers the time-span 1805-2015 and consists of nine earthquakes, whose causative faults are clearly recognised through extensive evidence of coseismic surface faulting (Azzaro, 1999). Earthquakes and associated faults used in the analysis are indicated in Figs. 1 and 2. Thus, the long-term mean recurrence time of historical major events in the Timpe area, reconstructed over a period of 210 years by the fault seismic histories (see Azzaro et al., 2013b), is just 23 years.

## 3.1 Characteristic magnitude and mean recurrence times by historical approach

Supported by the observation that major earthquakes have produced surface faulting ruptures along strike for the entire or most of the length of their causative faults (Azzaro, 1999), we assume that seismogenic Timpe faults behave according to the characteristic earthquake model (*sensu* Schwartz and Coppersmith, 1984). The earthquake size beyond which the phenomenon becomes evident corresponds to events having $I_0 \geq VIII$ EMS, equivalent to $M_W \geq 4.6$. These characteristic earthquakes therefore represent the maximum or quasi-maximum historically observed events: in Section 5.3 we will deal with the problem of maximum potential earthquake on faults by means of magnitude-size vs fault dimension relationships.

In previous studies we have calculated the mean recurrence time ($T_{mean}$) of a characteristic earthquake by simple intertimes statistics, given by the sum of all the inter-event times of major events divided by the number of the intertimes (see details in Azzaro et al., 2012b). Since the main goal is to bring the process of earthquake occurrence back to the scale of the individual fault, we calculated intertimes of earthquakes occurring on the same fault (in all, six intertimes), then we applied statistics obtaining a $T_{mean}$ of 71.3 years and an aperiodicity factor $\alpha=\sigma/T_{mean}=0.42$, a typical value for semi-periodic processes. Of course, in this way we assume that all the considered faults are characterised by the same values of $T_{mean}$ and $\alpha$ (Tab. 1). Given the intertimes dataset is not robust from a statistical point of view, we also applied a bootstrap analysis by sampling the initial IET dataset with replacement 1000 times to verify the confidence intervals of the results, similarly to the procedure adopted for paleoseismic datasets (Parsons, 2008), which typically are as "poor" as our sample. As a result, $T_{mean}$ remains stable while $\alpha$ is 15% lower than the value reported above.

The inset of Fig. 2 represents the retrospective analysis obtained by Azzaro et al. (2012b); the probability of having a characteristic earthquake on an individual fault in the next 5 years, a period chosen as representative of short-term earthquake rupture forecast in a high seismic rate region like Etna, is plotted versus time; the time invariant probability according to a Poisson distribution is represented by the horizontal pink line (at about 7% in 5 years), whilst the waves represent fault time-dependent probabilities calculated according to a Brownian Passage Time distribution (Matthews et al., 2002). The renewal process causes a sharp drop of the conditional probability function at the occurrence time of earthquakes assigned to each fault; note that all the historical events have occurred when the time-dependent probability of having an earthquake in the next 5 years is higher than the one derived with stationary assumptions, thus supporting the choice of time-dependency in our analysis. By doing this, we of course consider that fault behaviour inside SZ Timpe is somehow uniform, being affected by the same seismotectonic regime (Alparone et al., 2011).

## 4 Recent earthquake dataset: from the instrumental catalogue to the characterisation of seismic sources

Regarding short-term seismicity, we used data recorded by the seismic network of eastern Sicily that is operated by the Istituto Nazionale di Geofisica e Vulcanologia, Osservatorio Etneo, in Catania. Although the instrumental data at Etna have been collected since the early 1990s, a revised and complete earthquake catalogue has been compiled from 2000 by using a one-dimensional $V_P$ velocity model (Alparone et al., 2015; Gruppo Analisi Dati Sismici, 2016). For this study, we

considered only the portion of the catalogue from 2005 to 2015. In this time-window, the seismic release is generally regular both in terms of energy and numbers of events, not altered by the significant steps typically related with the seismic swarms accompanying eruptions at Etna, as occurred in 2001 and 2002-03 (Fig. 3). Moreover, since 2005 the seismic network has undergone a major upgrade, both in the number of stations and technology, with 3-component broad band seismometers and digital acquisition. This technological development has allowed detecting very low energy events (magnitude $\leq$ 1), the calculation of homogeneous and well-calibrated local magnitudes (Tuvè et al., 2015) and the application of advanced techniques for locating hypocentres (Mostaccio et al., 2013).

In order to better define seismic clusters or hypocentral alignments, thus contributing to the seismic source identification needed by the V3 Project, the 2005-2015 earthquake dataset was re-processed (Cocina et al., 2016) by using a three-dimensional $V_P$ velocity model (Alparone et al., 2012) and the tomoDDPS algorithm (Zhang et al., 2009). Compared to more simple methods, this code uses a combination of both absolute and differential arrival time readings between events of an earthquake cluster, so that for earthquakes with foci lying close to each other, travel time errors due to incorrect velocity models in the volume outside the cluster, are essentially ruled out.

As a result, we obtained a revised dataset consisting of 4,286 seismic events with $M_W$ up to 4.8; the magnitude of completeness of the catalogue $M_c$ is 1.1. Regarding the magnitude scale, the $M_W$ values of major recent earthquakes are taken from the literature or MedNet bulletin (http://mednet.rm.ingv.it/earthquakes.php), whereas we adopted the $M_L$-$M_W$ relationship calibrated on moment tensor analysis (Saraò et al., 2016) to convert the $M_L$ values reported in the catalogue. In general, most of shallowest earthquakes occurring at Etna in the 2005-2015 period are located in the eastern sector of the volcano within 7 km of depth (orange in Fig. 4a), clustering around the tectonic features of the Timpe and Pernicana fault systems. It should be noted that this seismicity is strictly related with the continuous fault activity and volcano-tectonic dynamics as a whole (Patanè et al., 2004; Solaro et al., 2010). Conversely, seismicity occurring at deeper crustal levels mainly represents purely tectonic regional dynamics due to the current compressive regime at the front of the Sicilian Chain-Foreland (La Vecchia et al., 2007; De Guidi et al., 2015; Scarfì et al., 2016). The most significant seismogenic volume in the deep crust beneath Etna is the one in the northwestern sector of the volcano, with focal depths in the range of 22- 30 km.

## 4.1 Area seismic sources

The area sources represent the most simplified representation of the fault systems that are relevant for seismic hazard. Area sources, or seismogenic zones (hereinafter SZ), are polygons including one or more faults where the earthquake occurrence rate is uniformly distributed and seismicity occurs at a defined (i.e. fixed) level of depth. This conceptual approach has been used in the past for the Italian seismic hazard map MPS04 (Meletti et al., 2008; Stucchi et al., 2011) and, more recently, for the European hazard map in the SHARE project too (Woessner et al., 2015).

Despite the detailed knowledge of the geometries of the active faults at Etna (Azzaro et al., 2013a), defining a SZ is not an easy task since the individual tectonic elements considered here are very close to each other, just 1 km apart in the case of the Timpe fault system (Azzaro et al., 2012a). The borders of the SZs are then defined as buffer zones around the fault lines

containing only the shallowest events occurring within 7 km depth (orange in Fig. 4a) of the relocated instrumental earthquake dataset. This is in agreement with the superficial nature of the volcano-tectonic structures, not rooted in the crust. In addition, we also grouped adjacent structures. In this way, we obtained four areal seismic sources – three for the Timpe system and one for the Pernicana system (blue polygons in Fig. 4a) – respecting the homogeneity in terms of other seismological and geological features ($M_{max}$, length and width, kinematics, slip-rate, see also De Guidi et al., 2012).

These SZs represent the recent seismotectonic activity of the shallowest crust (≤7 km) at Mt Etna, but also include all the strongest historical earthquakes ($M_W \geq 4.6$) associated with faults discussed before. About 1,000 earthquakes were used for the detailed characterisation of the areal sources. For an additional exploration on the epistemic uncertainties in defining source geometry, we also considered an extended SZ embracing the whole Timpe system, shown as a red polygon in Fig. 4a. In the following, we reported some graphs for the whole Timpe area; even if they are not used in the hazard computation

(see Peruzza et al., 2017), we believe they provide the reader with an insight on the uncertainties associated with the source geometry when a less detailed characterization mediating non-homogeneous behaviour inside the zone is used.

### 4.1.1 Effective depth

The characterization of the area sources includes the estimation of the effective depth, i.e. the seismogenic layer where most of the seismic energy is released. To this end, we calculated, by using the events included in each SZ, the distribution of the

number of earthquakes above the completeness threshold and the related strain release vs. the focal depth, with steps of 1 km. Results in Fig. 5 indicate that the seismogenic thickness is mainly confined to the first 5 km of crust, a value in agreement with the focal depth distribution of overall seismicity in the Mt Etna region (Fig. 4b). Note that, due to the cone-shaped topography of the volcano rising up 3000 m, hypocenters can be located above sea level (depth in these cases assumes negative values). In more detail, a first seismogenic layer can be observed at 0-2 km below sea level (bsl) in all SZs,

but also a second layer is evident at 4-5 km bsl defining the bottom of STF-SVF and MF-SLF area sources. It should be noted that major seismicity ($M \geq 3.0$ eqs.) occurs within both the layers (dark blue in Fig. 5). A similar pattern also emerges for the Timpe SZ, which includes the aforementioned individual SZs (except PF), confirming the main contribution to seismogenesis of the deeper focal depth level. In conclusion, SZs at Mt Etna are characterized by shallow effective depths, with PF and FF in the range 0 to 2 km and other sources between 0 to 5 km bsl (marked by orange stripes in Fig. 5). These

intervals are used as reference depths in the hazard computation (see details in Peruzza et al., 2017).

### 4.1.2 Seismic rate

Seismic rates have been determined by analysing the frequency-magnitude distribution from the instrumental earthquake catalogue by using the ZMap tools (Wiemer, 2001). The FMD of each SZ is estimated by maximum likelihood method (Wiemer and Wyss, 2002) using only the shallowest events (those occurring within a depth of 7 km bsl), so that $a$ and $b$

coefficients of the Gutenberg-Richter (GR) relationship are representative of the seismic activity of shallow sources. The magnitude of completeness $M_c$ of this subset of data is 1.3-1.4. The obtained FMDs (red in Fig. 6) indicate that the Timpe

faults (FF, STF-SVF, MF-SLF) have *b*-values varying from 0.84 to 1.13 (Tab. 2), while the Pernicana fault (PF) is characterized by a lower *b*-value (0.64).

To check if the FMDs obtained from an instrumental earthquake dataset during an interseismic period of just 11 years may represent the deformation processes driving the volcano-tectonic activity on the Mt Etna's flanks, and thus adequate to describe the long-term seismogenic behaviour, we calculated FMDs from the historical macroseismic catalogue (blue symbols in Fig. 6). The historical catalogue covers a time-span of ca. 150 years for all the SZs except for PF, whose anthropisation (and thus the seismic history) is limited to the last decades at most. Since the time extension of the instrumental and historical sub-catalogues is different, all the FMDs are represented after a normalization to one year. The visual comparison of the observed rates shows a satisfying match between macroseismic and instrumental data; there are no jumps or huge variations in slope, as often happens when dealing with such analyses, for example due to non-uniform magnitude assessment. For the Timpe sources (treated as a group, or separated in main fault systems in FF, STF-SVF, MF-SLF) the macroseismic FMDs are within the uncertainties of the instrumental ones, starting approximately above $M_W$ 3.5. Above this point, historical data represent the GR relationships for the high magnitudes, obviously not represented during an interseismic phase; conversely, the macroseismic FMDs deviate from the GR fitting at low magnitudes, thus representing the incompleteness of historical records for small earthquakes, a fact that it is widely known. Regarding the PF, the minor fit of instrumental and macroseismic FMDs is certainly due to the incompleteness of the macroseismic catalogue (short seismic history and events 'lost' because the area is largely uninhabited). Finally, we calculated α from GR according to Zöller et al. (2008) (see Tab. 1).

In conclusion, since we believe that the FMDs from instrumental and historical macroseismic catalogues match fairly well, we accept the simplification of adopting the 2005-2015 instrumental seismicity rates as proxies for the long-term seismogenic behaviour of area sources. Therefore *a* and *b*-values are calculated from the instrumental seismicity detected by high-quality monitoring during an interseismic period (i.e. in which no seismic swarm due to eruptions or volcanic activity has significantly affected our SZs) and will be used for characterizing the seismicity rates and extrapolating the GRs beyond the maximum value observed in these 11 years. The maximum magnitude has to be fixed on independent criteria that will be historical and/or geological, as described in the following.

## 4.2 Distributed seismicity

An alternative gridded seismicity approach has been used to depict 3D point sources in a crustal volume beneath Mt Etna. After several sensitivity tests, we calculated the *a*- and *b*-values of the GR relationship as follows: we created a three-dimensional grid with an inter-nodal distance of 2 km and applied a constant search radius of 3 km to sample the 2005-2015 instrumental earthquake dataset; grid nodes with less than 20 earthquakes were discarded. The maximum-likelihood method according to Wiemer and Wyss (2002) was applied for GR interpolation of events above the $M_c$ threshold (1.3); *a*-values have been normalized according to the volume represented. In this way, we obtained a grid consisting of 422 nodes; however, since the obtained sample of *a*- and *b*-values features scattering, we filtered the dataset by removing the outliers

and considered only the values between the 25[th] and 75[th] percentiles (Fig. 7a-b). As a result, the number of grid nodes used to characterise distributed seismicity is 359 (Fig. 7c).

We considered only the spatial variation of the *b*-value since the number of earthquakes in the grid nodes is not generally sufficient to be split into different time windows. Figure 8 shows the variability of the *b*-values at different depths beneath the Etna region. Variations are noteworthy in the first 7 km of the crust, with low *b*-values (≤0.8) characterizing the northern sector of the volcano around PF at very shallow levels of -2/-1 km, and higher *b*-values (≥1.2) in the central sector of Etna at a depth of 4 km. Note that in the eastern sector including the SZ Timpe at depths ranging from 2 to 6 km bsl, the *b*-value pattern varies widely both in value (0.7-1.2) and in space (patches extending a few kilometres). Finally, a relatively minor variation of *b*-values (0.9-1.1) is evident at intermediate crustal levels in the range of 10-16 km, while at depths higher than 20 km low *b*-values (≤0.9) prevail again.

This overall picture shows analogies with the pattern found by Murru et al. (1999, 2007) on a temporally different earthquake dataset (1999-2005), highlighting two areas characterised by higher *b*-values than other surrounding areas: i) beneath the central craters and ii) in the eastern flank, at a depth range of 5-7 km. Although the *b*-values cannot be compared as absolute numbers because they were calculated from two different magnitude scales - $M_L$ in this study, $M_d$ in the Murru et al. (2007) paper - the aforementioned spatial variations remain constant during time though the datasets cover contiguous time windows. The spatial distribution of the statistical parameters obtained from the IET analysis (Sicali et al., 2014) also displays similar lateral variations, indicating that the characteristics of earthquake occurrence in the central sector are very different from PF or SZ Timpe, the latter being more similar to the IET distribution observed in purely tectonic settings.

In conclusion, even if we cannot rule out that transient properties of the state of stress may influence the *b*-value in some cells, we believe that the above comparisons - as well as the overall good match between short term instrumental and historical catalogue seismicity rates (respectively red and blue dots in Fig. 6) - are sufficient evidence that seismicity rates deduced from a few years of instrumental seismicity during an interseismic period are representative of the longer term seismicity rates. They can thus be considered to represent the distributed seismicity in the source model.

## 5 Individual sources: seismic rates from geometric-kinematic fault parameters

In the previous chapters, seismic rates assigned to faults and area sources have been defined by historical-macroseismic and instrumental earthquake data. Taking advantage of the huge amount of geological field data and active tectonics evidence, we also performed a fault source modelling. This is based on a purely geological approach by converting the geometric-kinematic parameters representing fault activity into a budget of seismic moment potentially released by the structure through a computational scheme that also accounts for a magnitude-size scaling relationship (MSR). For each fault, we then obtain the most probable value of expected characteristic magnitude ($M_{char}$) with the associated standard deviation σ, the corresponding mean recurrence time ($T_{mean}$) and the aperiodicity factor α, which are the basic ingredients to compute earthquake occurrence probabilities, both under a Poissonian assumption as well as in a time-dependent perspective.

## 5.1 Method and input data

The analysis has been carried out using the software *FiSH*, a Matlab® routine developed to quantify the seismic activity of a fault from its geometric-kinematic parameters (Pace et al., 2016). The adopted approach is an evolution of the one by Peruzza et al. (2010) based on the criterion of 'segment seismic moment conservation' (Field et al., 1999). It takes into account the formal propagation of uncertainties in magnitude and slip-rate and uses directly the 3D fault geometry (length, dip-angle, thickness of the seismogenic layer) and slip-rate of a seismogenic structure. If a fault has a list of events associated to it, the mean values (magnitude, recurrence time) and their variability derive directly from historical or paleoseismological observations. However, there are very few cases of effective repetition of major earthquakes on the same fault segment in Italy, mostly along the Apennines in Central Italy (Galli et al., 2010; Cinti et al., 2011; Moro et al., 2013; Peruzza et al., 2011). At Mt Etna, some ten major earthquakes ($M_W$ 4.3-5.2) occurred repeatedly along the fault segments of the Timpe and Pernicana systems (Azzaro et al., 2012b).

The *FiSH* code uses different empirical and analytical relationships available in the literature between fault geometry and the characteristics of the expected earthquake, in order to quantify several values of $M_{max}$ and associated $T_{mean}$. Taking uncertainties of magnitude and slip-rate into account, the software provides budgeting of the seismic moment-rate. Finally, it uses the selected values to calculate the hazard rates, for a given exposure time, according to a Poissonian distribution or, in a time-dependent perspective that also considers the time elapsed since the last event, using some other widely used probability density function. For this study, the Brownian Passage Time (BPT, Matthews et al., 2002) is adopted to represent time-dependency.

Regarding our input data, the geometry, slip-rate and kinematics of the fault segments are constrained by detailed geological/geomorphological field investigations (Azzaro et al., 2012a; D'Amato et al., 2017) and geodetic data, the latter providing information on the vertical extension of faults as well as short-term slip-rates (Azzaro et al., 2013a). The 3D model of the individual sources considered in our application is shown in Fig. 9, together with the related geometric-kinematic parameters.

## 5.2 Magnitude-size scaling relationship for volcano-tectonic events

The characterization of an earthquake scaling relationship, which is suitable for a volcanic domain such as Etna, is a key step for modelling the rupture extent of these low to moderate magnitude events. Whereas empirical relationships derived for tectonic domains are widely available in the literature for both worldwide applications and regional contexts, those calibrated for active volcanic areas are relatively few. Among these, Stirling et al. (2013) mentioned those developed for thin crust volcano-tectonic contexts (Mason, 1996; Wesnousky, 2008), and the one specifically derived for the Taupo volcanic zone in New Zealand (Villamor et al., 2001).

At Mt Etna, major shallow volcano-tectonic earthquakes produce surface faulting with end-to-end rupture lengths up to 6.5 km and vertical offsets up to 90 cm. Systematic historical investigations and recent observations have enabled compiling an

earthquake rupture catalogue that reports some fifty coseismic faulting events (Azzaro, 1999, 2004). In this analysis, we use the most reliable observations of this dataset (43 data points) to derive a magnitude-scaling relationship that is specific for the Etna region, calibrated in the range $M_W$ 2.8-5.2 (Fig. 10a).

In Fig. 10b, Mt. Etna MSR is plotted together with the ones available for tectonic and volcanic domains. Considering the approximations due to the use of different dimensional measurements – magnitude scales, rupture length vs. rupture area – and the limitation in extrapolating the fitting outside the original magnitude ranges, the comparison is quite satisfactory. We note a strong analogy with respect to the trend of the relationship suggested by Villamor et al. (2001) for the Taupo volcanic zone, although the Etna one is scaled by ca. one order of magnitude, whereas discrepancies are substantial for thin crust volcano-tectonic context relationships proposed by Mason (1996) and Wesnousky (2008). Also the set of worldwide relationships by Wells and Coppersmith (1994) based on rupture length (RLD, SRL) tends, at different degrees, to overestimate the earthquake magnitude.

These considerations suggestedusing both the Etna and Taupo MSRs to calculate the seismic rates of the individual sources; in this way we tend to minimize the epistemic uncertainty associated with them. However, the effective interval of extrapolation of Taupo MSR is narrow, since the length of faults to be used for estimating expected $M_{char}$ is mostly in the range 7-11 km, i.e. next to the lower part of the Taupo MSR (see Fig. 10, frame b, length in logarithm scale).

### 5.3 Maximum expected magnitude and related mean recurrence times

The *FiSH* code calculates the value of magnitudes expected for the full rupture of each fault by the above-defined empirical scaling relationships. In order to check the geometrical consistency of the sources, it also estimates a maximum magnitude ($M_{max}$) according to: i) the scalar seismic moment ($M_{Mo}$), by using the modified formulation of magnitude (IASPEI, 2005) and a constant strain drop value of $3 \times 10^{-5}$; ii) an additional constraint based on the aspect ratio relationships ($M_{AR}$) derived by Peruzza and Pace (2002). Fig. 11 shows probability curves of all the $M_{max}$ values derived so far, assuming that a normal distribution represents the associated uncertainty, with a symmetrical bell-shape distributed around the central value; the maximum historical observed magnitude ($M_{obs}$) is also reported using the standard deviation of M assigned in the earthquake catalogue. The dashed curve (SUM) represents the summation of the probability density functions, whereas the vertical black line indicates the central value of its Gaussian fit to be considered as the reference mean value ($M_{max}$), with the associated standard deviation ($\sigma M_{max}$) given by the horizontal dashed line (for details see Pace et al., 2016).

In general, the magnitude values calculated by the different relationships are not drastically different from each other if the wide uncertainty ranges are taken into account. Note that the $M_{max}$ values are consistent with the $M_{obs}$ for the simplest and best documented sources (FF, STF); in the cases of more complex structures (e.g. PF and SLF) that are characterised by coseismic slip and creeping alternating in space and even in time along strike (Azzaro, 2004), the maximum observed magnitude always lies in the range of full rupture magnitude minus one standard deviation (($M_{max}$- $\sigma M_{max}$).

The mean recurrence times ($T_{mean}$) associated to $M_{max}$ values are computed accounting for slip-rate values and related uncertainties, which are strongly dependent (see Fig. 9); resulting $T_{mean}$ vary from 22 to 166 years (Tab. 1). However, these

values cannot merely be compared with those resulting from the analysis of the historical earthquake dataset, representative of the entire SZ Timpe (Tab. 1). Finally, the aperiodicity factor α, defined as the standard deviation of the recurrence times over their mean, has been estimated by introducing the formal error propagation to take account of the uncertainties in $M_{max}$ and slip-rates and so to explore how these uncertainties affect the variability of $T_{mean}$.

## 6 The final source models

The three types of seismic sources described above are used in the final seismic hazard assessment following the conceptual scheme reported in Fig. 12; for further details of the computation, the reader can refer to the companion paper, Part II (Peruzza et al., 2017). In brief:

• area seismic sources (cfr. Fig. 4) are horizontal planar surfaces of distributed (uniform) seismicity that encompass the best-known seismogenic fault systems on the eastern and northern flanks of Mt Etna; *a*- and *b*-values are calibrated on instrumental seismicity, effective depths are estimated by the analyses of strain release profiles, and $M_{max}$ is based on historical earthquake data. These sources represent the so-called 'Level 1', the simplest one with no branches, a first evolution of the Poissonian model used by the current seismic hazard map of Italian regulation (MPS04, Stucchi et al., 2011), where the whole volcanic edifice was enveloped into a single polygonal area;

• fault sources (cfr. Fig. 9) representing major earthquakes (M>4.5), are combined with the areas in a more complex source model, namely 'Level 2'. Here, the faults are individually modelled in terms of 3D geometry based on tectonic field data and geodetic information; they are assumed to behave according to a characteristic earthquake model; background seismicity is represented by the area sources of Level 1, where only earthquakes between M=$M_{min}$ and $M_{max}$=4.5 are modelled. The logic tree is in this case represented by 4 branches, based on historical or geological parameterisation of characteristic earthquakes, and on Poisson or time-dependent assumptions on recurrence intervals;

• the most complex source model is 'Level 3' that combines fault sources as in Level 2 with point sources (cfr. Fig. 8) which are used to represent distributed (gridded, non-uniform) seismicity. We prefer this the model as it is less driven by subjectivity in source definition, though it is not free of problems or questionable choices.

These levels form alternative seismic source models, stated in order of increasing complexity, to represent the epistemic uncertainties.

## 7 Conclusive remarks

In this paper we tackled the problem of characterizing low magnitude, shallow seismic sources, capable of affecting the seismic hazard for short exposure times at Mt Etna, the largest active volcano in Europe. Usually the problem of ground shaking due to local superficial volcano-tectonic faults is discarded in favour of estimates based on large-scale regional crustal faults capable of generating strong earthquakes (M>6); in addition, other major threats related to the eruptive activity,

or to the flank instability (Acocella et al., 2013; Acocella and Puglisi, 2013), can be first-order priorities for land planning and risk mitigation actions. But on Mt Etna's slopes, several inhabited localities have been repeatedly and heavily damaged as a consequence of local earthquakes with M<5.5 that may be connected to the eruption phases or not. In the documented history, such damage occurred on average every 20-25 years, the last sequence being along the S. Venerina fault (SVF) in 2002. To tackle these issues, the Italian Department of Civil Defense (DPC) has funded two research programs on Mt Etna aimed at mitigating, among other risks, the seismic one(Acocella and Puglisi, 2010; Azzaro and De Rosa, 2016). In this framework we started to characterise, with different methodological approaches, shallow sources and finally to assess the seismic hazard at the local scale of the volcano (Azzaro et al., 2012b, 2013b, 2016; Peruzza et al., 2017). Some basic ideas have driven our analyses i) a few years of high quality seismic monitoring in an 'interseismic' period can be representative of the long-term seismic rates of faults: ii) fault size and slip rate can constrain the maximum magnitude and the seismic moment budgeting, and geologic-geodetic derived seismic rates must be coherent with historical and instrumental data. If such ideas are true, we can extend the modelling of seismic sources to the whole volcanic complex by addressing 'unknown' faults by distributed point sources. we are then no longer forced to use independent events (i.e. the declustered earthquake catalogue, assuring stationarity of the process), but can compute the probabilities of events for any magnitude-frequency distribution for a generalized non-Poisson model.

We focused our analyses on two main volcano-tectonic fault systems evaluated at the surface and by geophysical investigations. Table 2 reports an overview of the relevant parameters to be used as input data in the companion paper by Peruzza et al. (2017). The Pernicana fault (PF) is an E-W oriented, S-dipping system of brittle and creeping transtensional segments: very shallow instrumental seismicity (located very often above the sea level) depicts quite well the 3D geometry of this structure characterized by low $b$-values (<0.7). The Timpe system in the SE flank is a group of nearly vertical normal faults. Their deep geometry cannot be precisely detected even by the high-quality instrumental earthquake dataset available in recent years. Area seismic sources have been depicted with increasing detail by using space buffers around the surface trace faults. Taken as a whole, the FMD of the SZ Timpe - as derived from the instrumental dataset of 2005-2015, a period that represents the 'interseismic background' level not affected by main earthquake sequences - is similar to the FMDs and depth distributions of the Moscarello-S. Leonardello faults (MF-SLF) , whilst the Fiandaca fault (FF) and S. Tecla-S. Venerina faults (STF-SVF) show, respectively, lower and higher $b$-values and activity rates. This apparent discrepancy can be accounted for by: i) the SZ Timpe also includes two small triangular areas (see upper right panel in Fig. 6), adding another 183 earthquakes (cfr. Fig. 4 caption); ii) the 'weight' of earthquakes of MF-SLF in terms of seismic moment released is much higher compared to the ones of FF and STF-SVF, and hence the similarity between SZ Timpe and MF-SLF is more evident.

Regarding the seismicity rates to be assigned to the faults, we note a global consistency by using the geometric-kinematic approach, and by using the historical earthquake dataset. The maximum magnitudes ($M_{max}$) calculated by scaling relationships appear ca. 0.3-0.6 units higher than the observed magnitudes ($M_{obs}$), whilst the related mean recurrence times ($T_{mean}$) are sometimes lower, modulated by the fast slip rates. There may be a number of reasons for these discrepancies,

such as: i) uncertainties of the geologic slip-rate estimations; ii) geometries of the modelled faults are not well constrained; iii) difficulty in discriminating pre- and post-seismic slip with respect the coseismic rupture length; iv) the role of fault segments in accommodating deformation (slip-rate partitioning).

Finally, the aperiodicity coefficients suggest sensitivity tests and care in modelling faults by a time-dependent approach: the $\alpha$'s obtained by geologic data indicate a quasi-stationary behaviour of the maximum-sized events, whilst the one calculated from the intertimes of historical earthquakes suggests a certain degree of periodicity. Both the seismicity rates for $M_{max}$, however, are within the uncertainties of rates derived by the GR relationships of instrumental data.

This work helps to improve our basic knowledge of seismogenic processes at Etna. Furthermore, it represents an effort to provide the international scientific community with original procedures and methodological approaches to produce hazards maps in other volcanic areas.

**Acknowledgements**

Many thanks are due to G. Weatherill and C. Beauval for their fruitful comments and constructive criticism. The editor O. Scotti is also acknowledged for her valuable suggestions. R. Gee gave useful hints for developing some of the numerical models in the initial stage of the work. This study has benefited from funding provided by the Italian Presidenza del Consiglio dei Ministri –Dipartimento della Protezione Civile (DPC), in the frame of the 2012-14 Agreement with Istituto Nazionale di Geofisica e Vulcanologia-INGV, Project V3 "Multi-disciplinary analysis of the relationships between tectonic structures and volcanic activity".

This paper does not necessarily represent DPC official opinion and policies.

The authors are grateful to S. Conway for revising the English text.

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

| Fault | Historical eqs dataset | | | | Instrumental eqs dataset | | Geological-kinematic | | | | | |
|---|---|---|---|---|---|---|---|---|---|---|---|---|
| | $M_{obs}$ | $\sigma M_{obs}$ | $T_{mean}$ (yr) | $\alpha$ | $\alpha$ | $M_{min}$ | $T_{mean}$ (yr) | $\alpha$ | $M_{max}$ | $\sigma M_{max}$ | $M_0$ rate (Nm) | $\mu$ (GPa) |
| Pernicana (PF) | 4.7 | 0.30 | | | 0.52 | 4.3 | 28 | 1.04 | 5.0 | 0.3 | 1.42E+15 | 11 |
| Fiandaca (FF) | 4.6 | 0.36 | 71 | | 0.62 | 4.3 | 166 | 1.04 | 4.9 | 0.3 | 1.70E+14 | 12 |
| S. Tecla (STF) | 5.2 | 0.36 | 71 | 0.36* / 0.42 | 0.78 | 4.3 | 53 | 1.08 | 5.3 | 0.3 | 2.12E+15 | 15 |
| S. Venerina (SVF) | 4.6 | 0.36 | 71 | | 0.78 | 4.3 | 45 | 1.05 | 5.0 | 0.3 | 8.85E+14 | 15 |
| Moscarello (MF) | 4.9 | 0.36 | 71 | | 0.66 | 4.3 | 119 | 1.42 | 5.5 | 0.4 | 1.88E+15 | 15 |
| S. Leonardello (SLF) | 4.0 | 0.36 | 71 | | 0.66 | 4.3 | 22 | 1.37 | 4.8 | 0.4 | 9.06E+14 | 15 |

**Table 1: Comparison with estimations based on historical and instrumental earthquake datasets; the explanation of the geological-kinematic approach is given in chapter 5. Abbreviations: $T_{mean}$, mean recurrence time; $\alpha$, aperiodicity factor; $M_{max}$, maximum magnitude obtained by the FiSH code and related standard deviation ($\sigma M_{max}$); $M_0$, moment rate; $\mu$, shear modulus; $M_{min}$, minimum magnitude of the instrumental earthquake dataset. Asterisk indicates the value obtained by the bootstrap analysis.**

| | PF | MF-SLF | STF-SVF | FF |
|---|---|---|---|---|
| Effective depth (km) | 0 to 2.0 | 0 to 5.0 | 0 to 5.0 | 0 to 2.0 |
| *b*-value | 0.64 ± 0.06 | 0.91 ± 0.08 | 1.13 ± 0.16 | 0.84 ± 0.15 |
| annual *a*-value | 2.08 | 2.51 | 2.73 | 1.72 |

**Table 2: Effective depth, b and a coefficients of the GR relationship for each SZ, obtained from the instrumental earthquake dataset (2005-2015).**

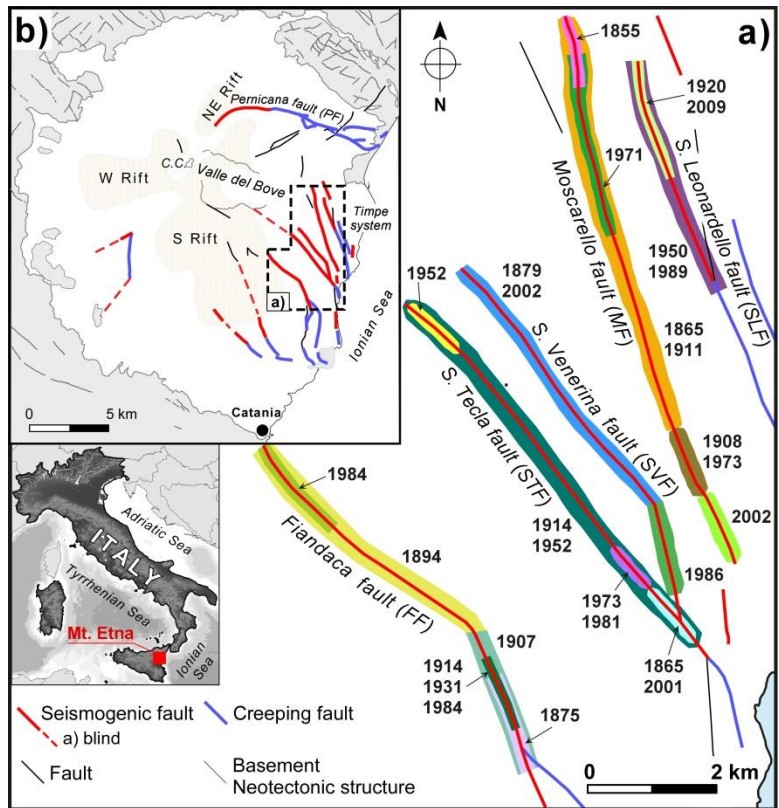

**Figure 1: Fault systems in the study area: a) Patterns of historical surface faulting along the southeastern flank of Mt Etna (Timpe tectonic system); colours represent coseismic ruptures related to different earthquakes (modified from Azzaro, 1999). b) Seismotectonic model of Etna (from Azzaro et al., 2012a). In beige the rift zones, i.e. high frequency of opening of eruptive fissures; in grey the sedimentary and metamorphic basement underlying the volcano.**

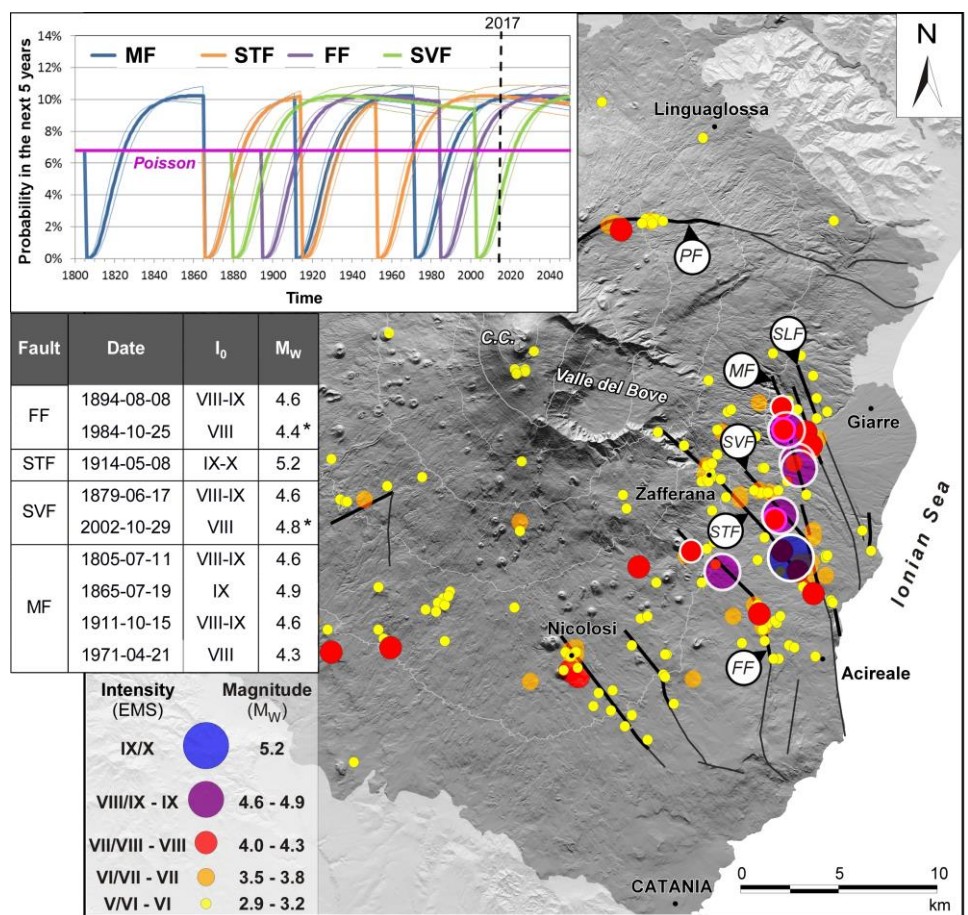

| Fault | Date | $I_0$ | $M_W$ |
|-------|------|-------|-------|
| FF | 1894-08-08 | VIII-IX | 4.6 |
| | 1984-10-25 | VIII | 4.4* |
| STF | 1914-05-08 | IX-X | 5.2 |
| SVF | 1879-06-17 | VIII-IX | 4.6 |
| | 2002-10-29 | VIII | 4.8* |
| MF | 1805-07-11 | VIII-IX | 4.6 |
| | 1865-07-19 | IX | 4.9 |
| | 1911-10-15 | VIII-IX | 4.6 |
| | 1971-04-21 | VIII | 4.3 |

| Intensity (EMS) | Magnitude ($M_W$) |
|-----------------|-------------------|
| IX/X | 5.2 |
| VIII/IX - IX | 4.6 - 4.9 |
| VII/VIII - VIII | 4.0 - 4.3 |
| VI/VII - VII | 3.5 - 3.8 |
| V/VI - VI | 2.9 - 3.2 |

Figure 2: Distribution of the historical seismicity in the Etna region from 1600 to 2015 (data from CMTE Working Group, 2017). Major events considered for the analysis are outlined by a white circle and listed in the enclosed table (asterisk indicates an instrumental value); fault pattern and abbreviations as in Fig. 1, C.C. indicates the central craters. Inset map shows the retrospective test of the time-dependent model based on intertimes and *b*-values of faults: colored curves indicate the variation in time of the conditional probability assigned to the faults of the SZ Timpe, in the next 5 years. Before the first event assigned to each fault, the probability is assumed as Poissonian; following the earthquake, the probability curve collapses and progressively increases until the next rupture (from Azzaro et al., 2013b).

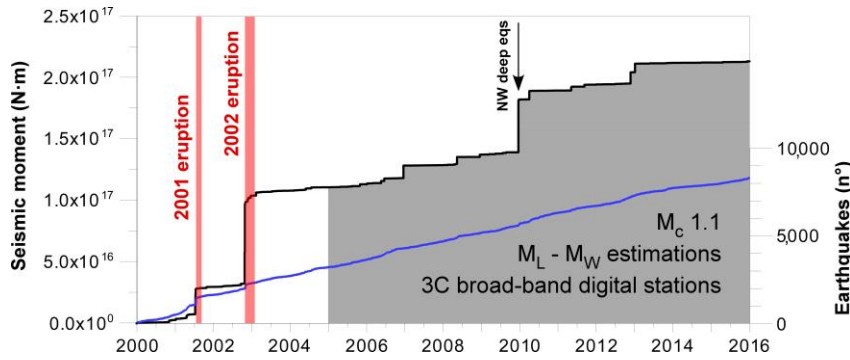

**Figure 3: General trend of seismicity at Mt Etna from 2000 to 2015: the black line indicates the cumulative seismic moment (calculated according to Kanamori, 1977), the blue line shows the number of earthquakes. The periods marked in red indicate the main flank eruptions; in grey, the time-span selected for the analysis. Note that the step in the seismic release at the end of 2009 is related to a seismic sequence in the NW sector of the volcano at a depth of 24-28 km, not affecting the characterization of the shallow sources.**

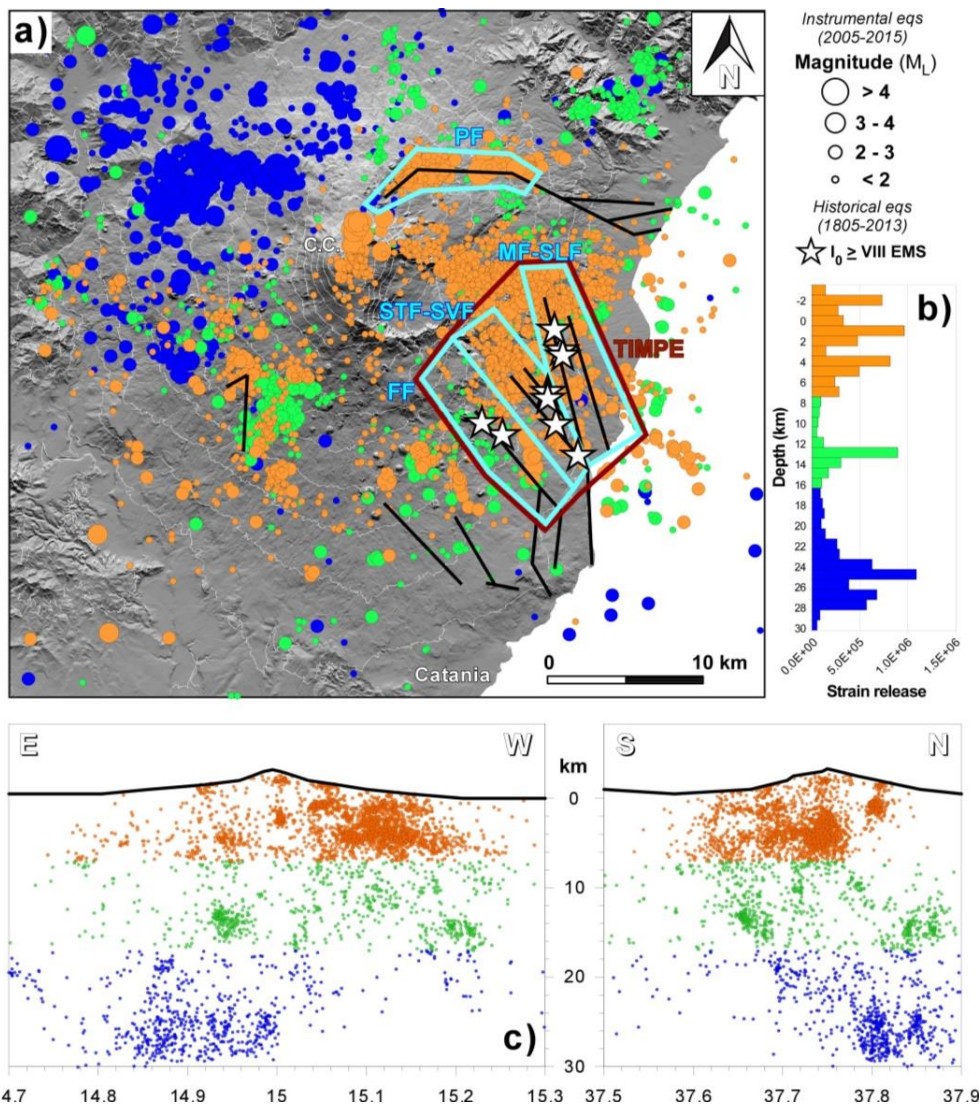

**Figure 4: a)** Historical and instrumental seismicity used for characterizing seismic sources at Etna. Areas in light blue indicate the seismic zones: PF, Pernicana fault (295 eqs.); MF-SLF, Moscarello and S. Leonardello faults (354 eqs.); STF-SVF, S. Tecla and S. Venerina faults (313 eqs.); FF, Fiandaca fault (69 eqs.); Timpe (919 eqs.). Solid black lines represent the simplified pattern of active faults. **b)** Distributions of seismic strain release vs. focal depth for the 2005-2015 instrumental earthquake dataset referring to the entire Etna region. **c)** Cross-sections of the 2005-2015 instrumental earthquakes beneath the volcano.

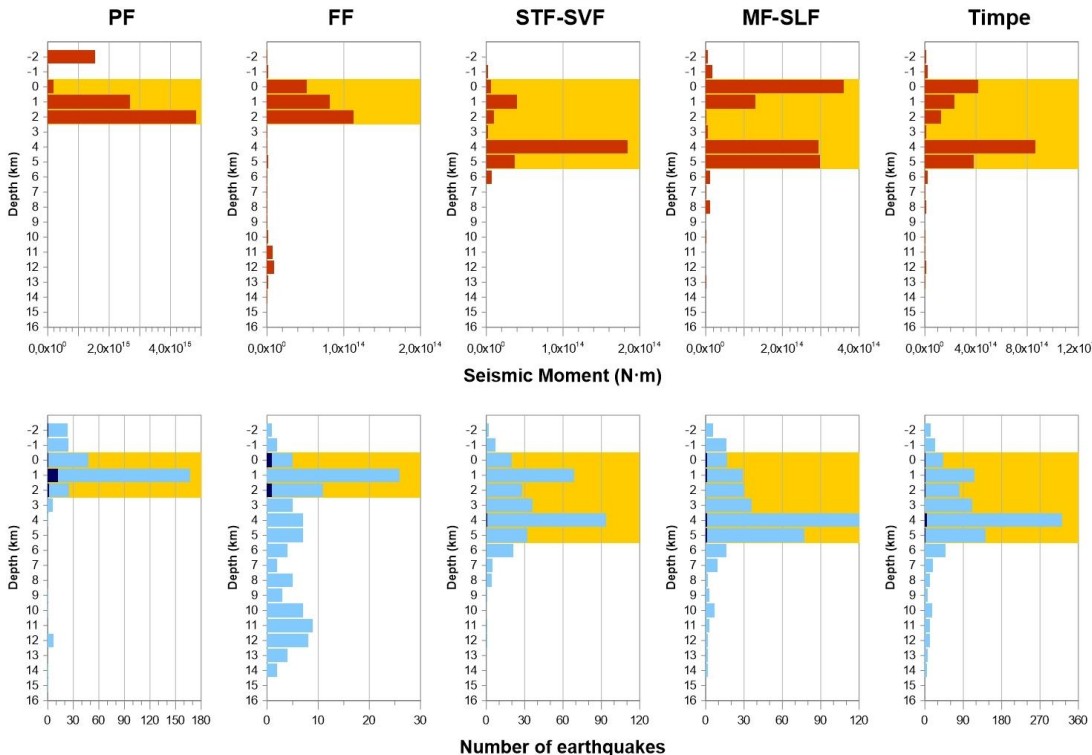

**Figure 5:** Distributions of seismic strain release (top) and number of earthquakes (bottom) vs. focal depth for the SZs considered in the model. Dark blue histograms indicate the number of earthquakes with $M_W \geq 3.0$. The effective depth is marked in orange. Abbreviations as in Fig. 4.

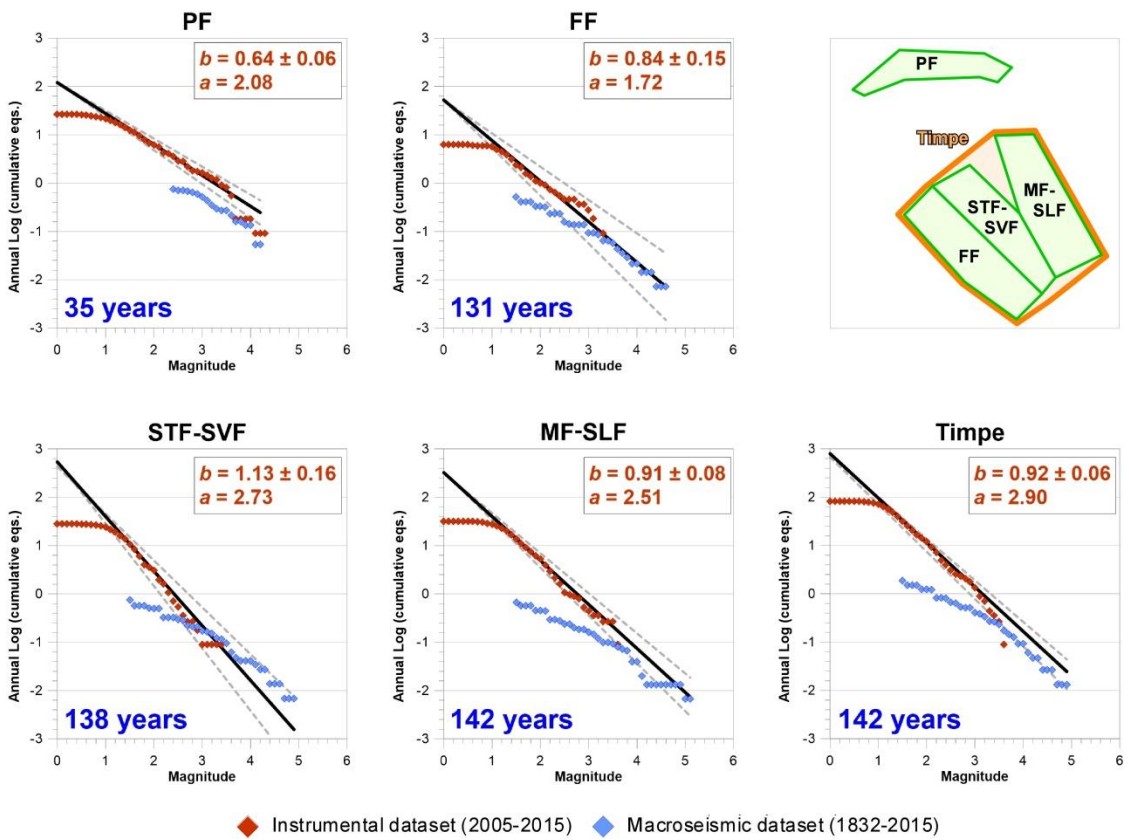

**Figure 6: Frequency Magnitude Distribution for each SZ. Red dots refer to the instrumental dataset, blue dots to the macroseismic one; dotted lines indicate uncertainties concerning the GR relationship (black line). *b*- and *a*-values are obtained from the instrumental earthquake dataset. Years indicate the actual time-window ($T_{last}$-$T_{first}$) of the events in each sub-catalogue of the historical dataset. Data are normalized to one year. Abbreviations as in Fig. 4.**

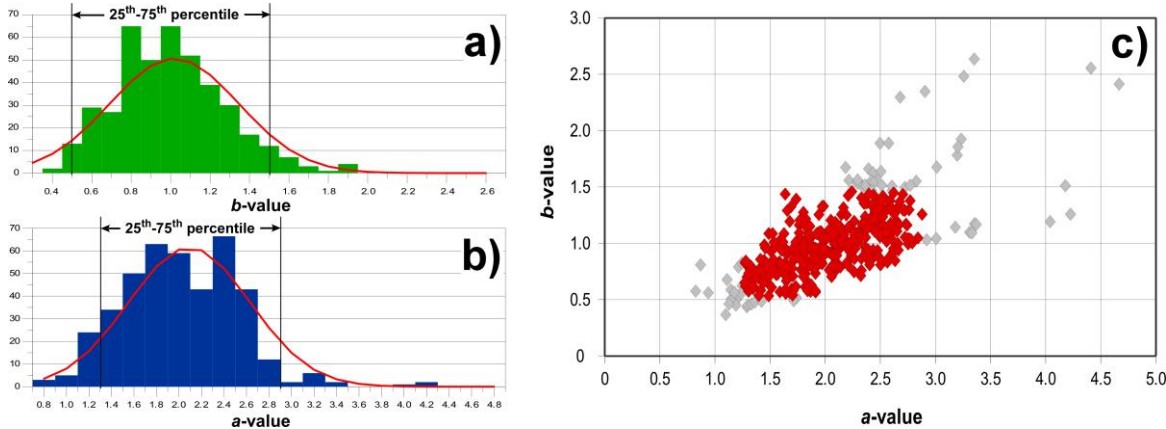

**Figure 7: a-b) Histograms showing the frequency distribution of *b*- and *a*-values. c) Plot of *a*- and *b*-values obtained for the grid nodes; discarded values are in grey.**

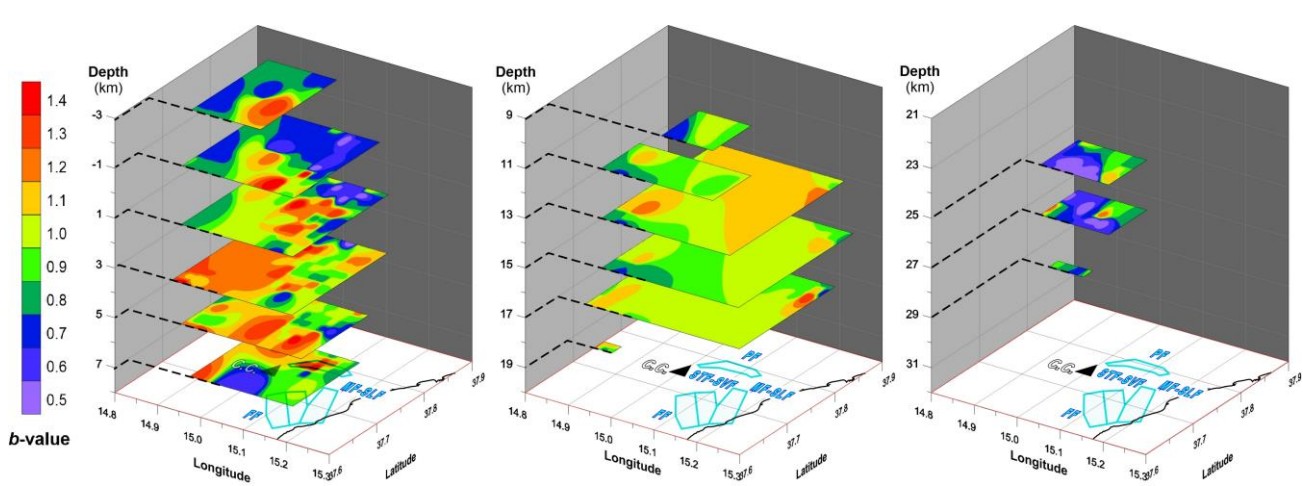

**Figure 8: Distribution of the *b*-values beneath Etna calculated from the instrumental earthquake catalogue (2005-2015): the horizontal sections show the grids at different depths.**

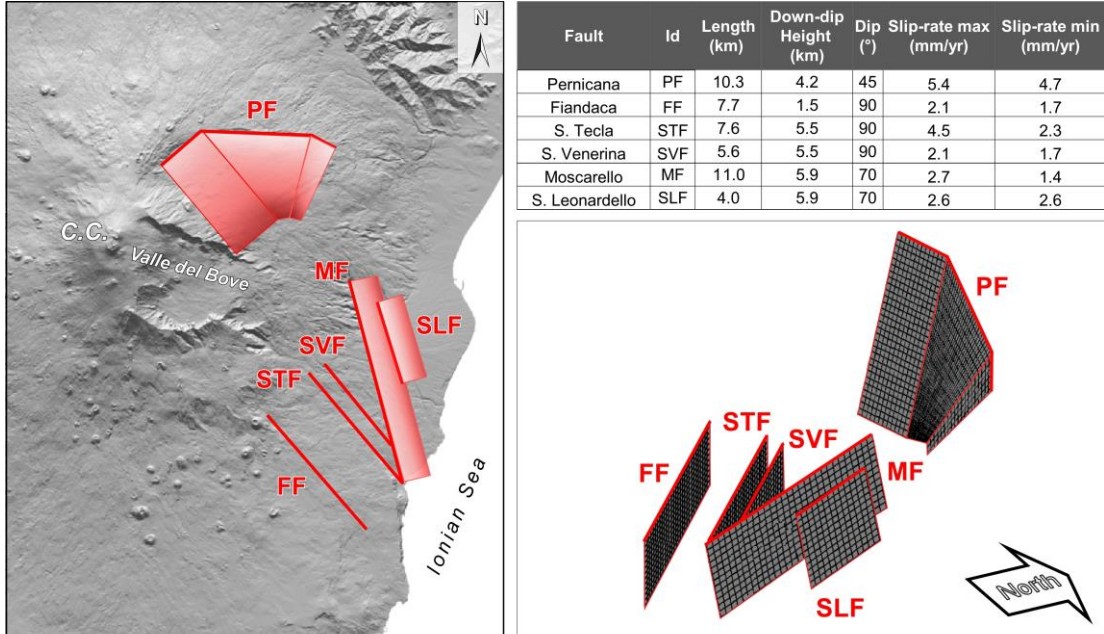

| Fault | Id | Length (km) | Down-dip Height (km) | Dip (°) | Slip-rate max (mm/yr) | Slip-rate min (mm/yr) |
|---|---|---|---|---|---|---|
| Pernicana | PF | 10.3 | 4.2 | 45 | 5.4 | 4.7 |
| Fiandaca | FF | 7.7 | 1.5 | 90 | 2.1 | 1.7 |
| S. Tecla | STF | 7.6 | 5.5 | 90 | 4.5 | 2.3 |
| S. Venerina | SVF | 5.6 | 5.5 | 90 | 2.1 | 1.7 |
| Moscarello | MF | 11.0 | 5.9 | 70 | 2.7 | 1.4 |
| S. Leonardello | SLF | 4.0 | 5.9 | 70 | 2.6 | 2.6 |

**Figure 9: Pattern of individual sources used in the geological model and related geometric-kinematic parameters: red boxes in the left frame represent the projection at the surface of the fault planes, lines indicate the vertical planes. Note that lengths refer to the seismogenic fault segments only, whereas the ones governed by prevailingly creeping behaviour are not considered.**

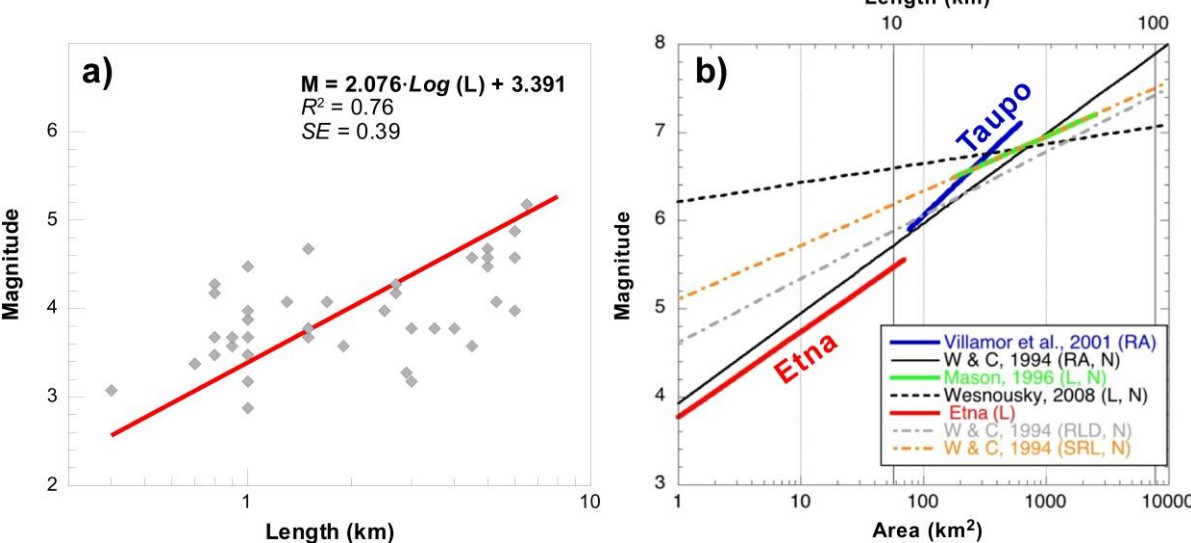

**Figure 10: a) Plot of earthquake magnitude vs rupture length for the Etna region (this study); b) Comparison with the magnitude-size scaling relationships for the Taupo volcanic zone (Villamor et al., 2001) and other relationships worldwide (Wells and Coppersmith, 1994; Mason, 1996; Wesnousky, 2008). Abbreviations: L, fault length; N, normal kinematics; RA, rupture area; RLD, rupture length at depth; SRL, surface rupture length.**

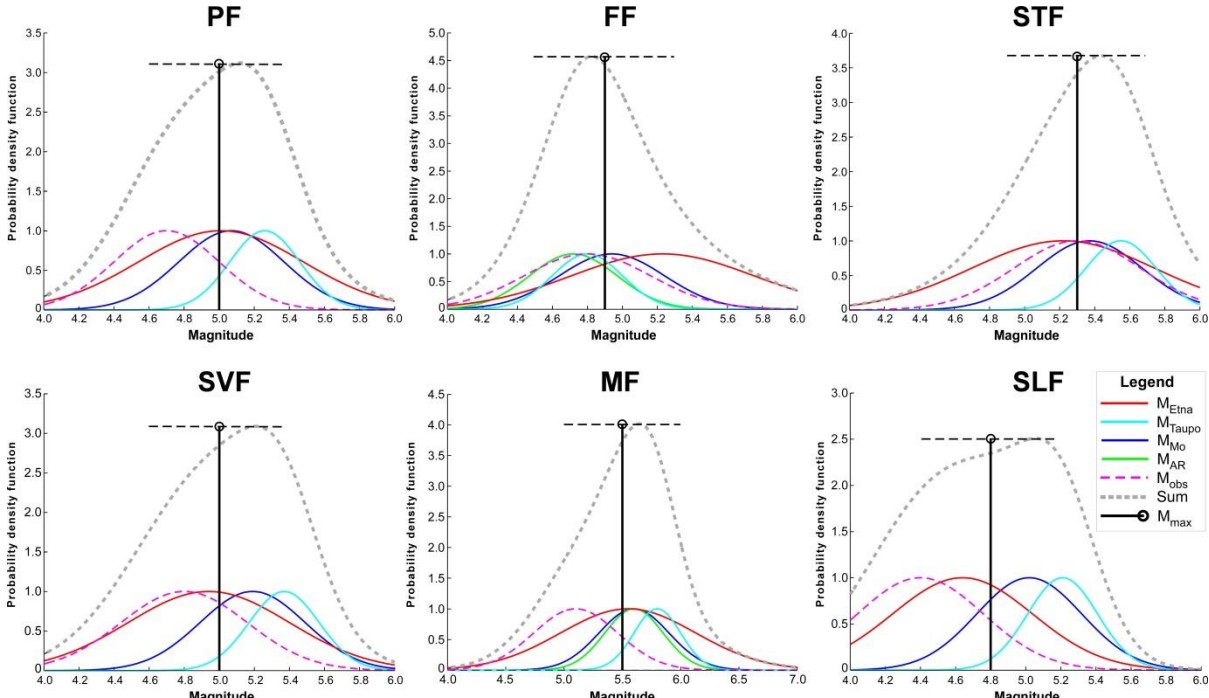

**Figure 11: Maximum magnitudes ($M_{max}$) estimated by the *FiSH* code for the studied faults. Abbreviations: $M_{Etna}$-$M_{Taupo}$, magnitude from earthquake scaling relationships for Etna and Taupo; $M_{Mo}$, scalar seismic moment magnitude; $M_{AR}$, magnitude from aspect ratio relationships; $M_{obs}$, maximum observed magnitude. Uncertainties are represented by probability curves (see text for explanation).**

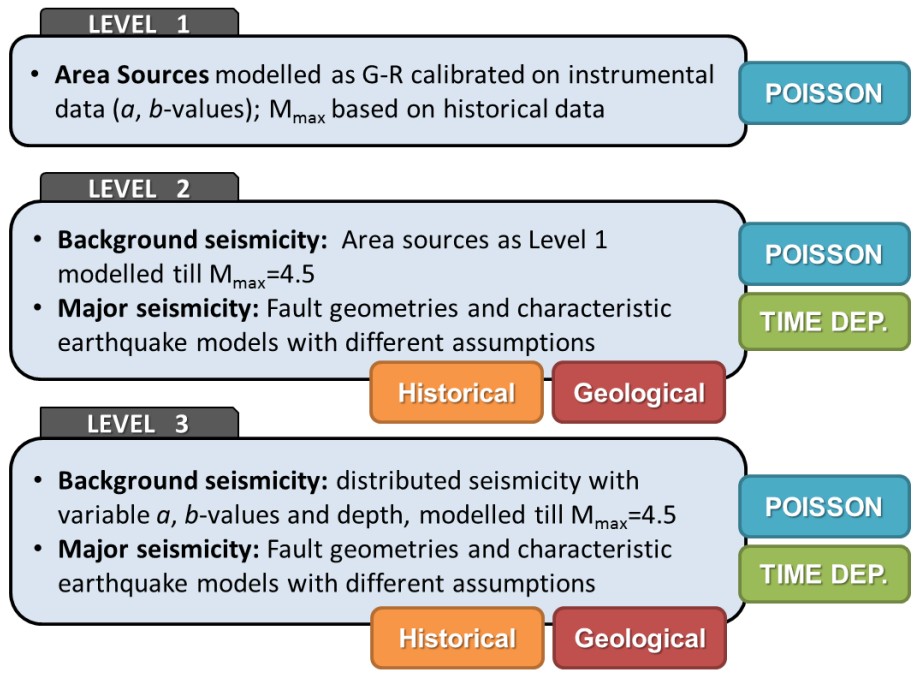

**Figure 12: Schematic chart describing the three levels of the source models defined for the Mt Etna region: increasing complexities are introduced from Level 1 to 3; the final logic tree we adopt is Level 3, with four branching levels. Details of computations are given in Part II (Peruzza et al., 2017).**