# Peer review of "When probabilistic seismic hazard *climbs* volcanoes: the Mt Etna case, Italy. Part I: model components for sources parametrization"

_Natural Hazards and Earth System Sciences, 2017_

## Referee Comment (RC1) · G. Weatherill (Referee) · 11 May 2017

The manuscript describes the construction and characteristics of seismogenic source models for use in probabilistic seismic hazard analyses (PSHA) in the Mt Etna region. The models are derived from short-term instrumental seismicity, long-term historical seismicity and from the fault geology. Volcanic regions such Mt Etna pose an important challenge in PSHA modelling due not only to the complexity of the source process, but also to the very nature of the forces controlling the seismicity, which may be by-definition non-Poissonian in nature. As a practical approach, illustrating the possible means by which a seismogenic source model can be constructed in such a region using available seismic and geological data, the paper is a valuable contribution to the topic.

It is easy to see how it can be incorporated into models in other volcanic environments. The specific methodologies within the approaches are generally applied in a sound manner and build upon the current state-of-practice in PSHA source modelling. On this basis, I believe the paper represents a solid contribution to the scientific literature and should be considered for publication within this journal.

There are, however, some specific topics where the material presented seems somewhat incomplete. The main topic omitted in this paper is the characterization of epistemic uncertainty in the source model and the manner in which this is formulated for a PSHA calculation. Whilst some discussion on this topic can be found, albeit briefly, in section 6 of the accompanying Peruzza et al. manuscript, complete omission of the epistemic uncertainties and the combination of the different model approaches presented here diminishes the value of this manuscript as a stand-alone paper on source modelling in volcanic regions. I recommend the authors to consider adding some additional discussion here as to how epistemic uncertainty should be treated and outline the basic formulation of the logic tree. Some overlap with Peruzza et al. (2017) is tolerable in this case.

Secondly, the authors explain in section 2 that destructive historic events have occurred both in periods of activity as well as times of quiescence, and that the recurrence models for slip in larger characteristic events behave in a manner typical of those under tectonic stress rather than local magmatic stress. Whilst it is possible to accept this at face value given the lack of correlation mentioned, it is not so true to say that this applies to all the other seismicity. Within the distributed sources and area zones the recurrence is dependent upon the rate of all seismicity, which will be more closely linked to cycles of eruptive activity and quiescence. In a PSHA risk mitigation context, this means that if considering the probability of exceeding ground motion in a given time period (e.g. 5, 30, 50 years) one needs to account for the probability of an eruptive episode and the probability of exceeding the given levels of ground motion conditional upon the occurrence of the eruptive episode. This is in addition to the baseline hazard

during periods of quiescence. Of course, this widens discussion regarding the quantification of the probability of eruptive episodes, but it may be important for putting this work into practice. This is not a critical flaw in the methodology described, but may be a theoretical limitation of the assumptions made in the application of the recurrence models for the area and distributed seismicity sources.

For the manuscript in general, the writing is of a quality sufficient for publication. Some notable typing errors are indicated below, but I would recommend the authors to check further before compiling the final manuscript. Figures and tables are clear and well captioned.

Comments in Detail:

Lines 19-20: "We derive a magnitude-size scaling relationship specific for this volcanic area" – Change "specific for" to either "specific to" or "specifically for"

Lines 20 – 21: "Pace et al. (2015)" is "Pace et al. (2016)" in bibliography

Lines 25: "These analyses to not account regional M>6" – "Do not account for regional . . ."

Line 29: "However, apparently less evident . . .", can be changed to just "Less evident but equally . . ."

Line 36: comma needed after first "which"

Line 37: Should be "computation codes developed for the whole of Italy"

Line 62: comma needed after "widespread" and then removed after "eastern flank"

Line 82: ". . . seismic hazard applications regards the question of . . ." is better phrased as ". . . seismic hazard applications is the question of . . ."

Line 84: "It is a matter of fact that destructive earthquakes in the Tiempe area historically occurred both during flank eruptions and not" – Perhaps change the "and not" to

"as well as during periods of volcanic quiescence".

Lines 104 -105: "It has to be noted that moderate values of magnitude for heavily damaging events are a feature of seismicity in active volcanic areas such as Etna, whereas in tectonic domains crustal earthquakes producing the same effects are generally associated with M > 6." This comment has particularly significant implications for seismic hazard analysis and I would encourage the authors to: i) add a citation, ii) if known, briefly summarise what are believed to be the potential factors that may explain this observation.

Line 132: Needs comma after "somehow uniform"

Line 170: Replace "global" with "European".

Line 180: The use of the detailed areal sources and the extended sources are not clear. Are these alternative branches on an epistemic uncertainty analysis as the comment regarding uncertainty would apply? If so, then the authors need to clarify how the two different models are weighted. If not, then it is unclear how the authors are partitioning the moment rate between the two models.

Section 4.1.2: The assertion of a Gutenberg Richter model for the various faults is not entirely consistent with the observation shown in Figure 6. In nearly all cases the observed rate of earthquakes around M 3 is greater than that implied by the GR models, which suggests some kind of hybrid characteristic model. This may be shifting the trend toward lower b-values. Did the authors consider a hybrid model in which larger events occur more frequently than predicted by GR? The trend is less obvious for the Timpe zone, which reflects a common perception of GR-behavior across zones spanning larger spatial domains.

Section 4.2: The usage of distributed seismicity in this context should be debated more than is done so here. Given the relative brevity of the seismic catalogue, when looking at b-value variation on a fine spatial resolution it may be increasingly likely that the

values in any given cell may reflect a transient process. Even if the variation in b-value is cannot be attributed to statistical artefact, can the authors rule out the possibility that they are related to transient properties of the state of stress around particular elements in the complex volcanic system (including interaction with fluids), even if the period is quiescent? How representative might these values be of recurrence on a multi-decadal timescale?

---

## Referee Comment (RC2) · C. Beauval (Referee) · 22 May 2017

When probabilistic seismic hazard climbs volcanoes : the Mt Etna case, Italy. Part I : model components for sources parameterization

By R. Azzaro et al.

The paper presents a thorough work done to characterize earthquake recurrence at Mt Etna, to constrain the source model required for PSHA estimation. Earthquake recurrence is estimated using different datasets: historical seismicity, an instrumental catalog, as well as geological and geodetical deformation estimations. Mt Etna appears to be a well-studied zone, perfect for testing methods to estimate probabilistic seismic

hazard in the volcanic environment.

The manuscript is made of 3 sections: (1) historical data and estimation of mean recurrence times for earthquakes on main faults; (2) instrumental catalog and estimation of frequency-magnitude distribution for source zones enclosing faults, frequency-magnitude distributions estimated for gridded seismicity (instrumental catalog); (3) characteristic model for faults with parameters inferred from geological and geodetical slip rates.

The manuscript is well written and the content is very rich, but sometimes it lacks clarity. I am detailing below the issues that would need to be solved. There are also some technical questions that need to be addressed.

Main remarks

With a title announcing a PSHA framework, the reader expects a description of the final model used for probabilistic calculations. However, the authors do not explain how the different recurrence models will be combined to build final models, and how the source model logic tree will be built. Will the area zones and fault models constitute alternative models ? Is the gridded seismicity used as background for the fault model, and how? This information is essential to understand how the source model for PSHA is built.

Section 3, addressing earthquake recurrence from historical data, is a quick summary of a published paper (Azzaro et al. BGTA 2012). An interesting work has been done to estimate probabilities of occurrence of earthquakes on faults. However, as it is, it is not possible to fully understand the text and the results (inset figure in Fig. 2). Either this part has to be expanded, or it should be reduced and refer more strongly to the 2012 paper. Why the 1st case "events occurring everywhere inside the SZ Timpe" leads to "eight intertimes", and the second case "events occurring at the scale of individual faults" leads to "six intertimes"? The inset of Fig. 2, probabilities of occurrence of an earthquake in the next 5 years, refers to the 2nd case (+ why considering a 5 years period?)? Apparently, faults are assumed to have the same mean recurrence time, but

this is not explained. The inset is too small to correctly appreciate the curves. The time-dependent model, Brownian Passage Time model, should be introduced. The last sentence, referring to a bootstrap analysis, is difficult to understand without more explanations on the test done.

Section 4 describes the instrumental earthquake catalog, the delineation of area sources, the determination of seismogenic depths, the estimation of Gutenberg-Richter models for the area source zones, and the estimation of recurrence parameters for a gridded seismicity model. Area sources here are buffer zones around faults. Can you give more precisions on how this buffer zone is delineated (width, association of earthquakes with the fault)? Frequency-magnitude distributions based on the instrumental data is compared to the frequency distribution based on historical data. Why not combining both? E.g. at the scale of the Timpe zone, combining both would lead to a recurrence model fitting both the instrumental magnitudes (interval 2-3.5) and the historical magnitude rates (interval 3.5-5.0), instead of over-estimating slightly the historical rates? Magnitudes larger than 3.5 are contributing strongly in the probabilistic hazard estimation. Nothing is said about the presence of clustered events in the catalog (swarms, foreshocks, aftershocks?). How do the authors handle this issue, which is of importance when establishing earthquake recurrence models and calculating b-values?

Distributed seismicity : rather than arbitrarily excluding cells where "strange" b-values have been obtained, would it be possible to apply some criteria on the estimation of the b-value, e.g. increase the minimum number of events in the cell or impose a minimum magnitude range available? These criteria would ensure the reliability of the recurrence curve inside each cell. Besides, are the b-values obtained within the values expected for volcanic areas? It is hardy possible to locate the b-values mapped in Fig. 8, without any topography or country border.

The magnitude-size scaling relationship for the Taupo volcanic zone is compared to the relationship for Mt Etna, then both are used for estimating maximum magnitudes.

Is it correct to compare 2 relationships established on disjoint datasets (for Taupo, minimum length is larger than 10km, while for Etna maximum length is around 10km) ? Could you add a short discussion on extrapolating scaling relationships? The section concludes saying that "the mean recurrence times associated to Mmax values vary from 22 to 166 years, periods generally consistent with those historically observed for the individual faults". I don't understand the sentence, as the mean recurrence time estimated from the historical dataset is 71 years (Table 2). How is the aperiodicity factor estimated in this model (Table 2)?

Other remarks

Section 3.1 should be suppressed, as there is no Section 3.2.

Section 4, Figure 6: the time period used to estimate annual rates of historical earthquakes for SZ Timpe, FF, STF-SVF, MF-SLF, should be the same? As the completeness of historical data must be homogeneous within the rather small Timpe zone? The time period indicated in the legend, 1832-2015, corresponds to a larger time window, 184 years. Why reducing this time window to 138 or 142 years to calculate annual rates? Please make this point clear, as it is confusing.

L 248: "This overall picture is consistent with the inter-time distribution of earthquakes (Sicali et a; 2014)" : please, how do you relate b-values with inter-time distributions?

Table 2: should be cited in Section 3 dealing with mean recurrence times. Why providing the "Mmin for which is calculated the probability of occurrence" as it is not mentioned nor discussed in the text?

L184-185: "This option has a dual purpose: i) to provide a less detailed characterization mediating features inside heterogeneous, " => there must be a word missing?

L 295: "Considering the approximations due to the use of different dimensional measurements, the comparison is fairly explanatory" => exploratory?

L 321: See tab 1 : should be Table 2?

L350 [conclusion] "Taken as a whole, the FMD of the SZ Timpe is similar to the FMDs and depth distributions of the Moscarello (MF) and S. Leonardo faults (SLF), whilst the Fiandaca Fault (FF), S. Tecla and S; Venerina faults (SVF) show, respectively lower and higher b-values and activity rates." => sentence which is confusing and needs to be re-phrased. MF and SLF belong to the same SZ. Timpe encloses MF-SLF, FF, and STF-SVF, so the seismic rates in Timpe must be higher or equal to the sum of these 3 FMDs.

---

## Author Comment (AC1) · 28 Jul 2017

Dear Celine, thank you very much for the careful revision, we appreciated your comments and suggestions aimed at improving the manuscript by clarifying not fully explained passages. We answered to all the questions you raised and modified the text accordingly. Hereinafter the detailed list of your comments and our replies, a zip file containing the revised manuscript with tracked changes, and the new figures that have been modified. On behalf of all the authors Raffaele Azzaro

R2. Main remarks With a title announcing a PSHA framework, the reader expects a description of the final model used for probabilistic calculations. However, the authors do

not explain how the different recurrence models will be combined to build final models, and how the source model logic tree will be built. Will the area zones and fault models constitute alternative models ? Is the gridded seismicity used as background for the fault model, and how? This information is essential to understand how the source model for PSHA is built.

We agree with you, this is a point raised by the other reviewer too. We decided to accept some overlapping with Part II paper for having a more readable and stand-alone paper. We thus entered some new lines in the introduction and a new block of text now marked as Chapter 6, where we added a picture representing the logic tree approach (taken from Part II paper).

R2. Section 3 Section 3, addressing earthquake recurrence from historical data, is a quick summary of a published paper (Azzaro et al. BGTA 2012). An interesting work has been done to estimate probabilities of occurrence of earthquakes on faults. However, as it is, it is not possible to fully understand the text and the results (inset figure in Fig. 2). Either this part has to be expanded, or it should be reduced and refer more strongly to the 2012 paper. Why the 1st case "events occurring everywhere inside the SZ Timpe" leads to "eight intertimes", and the second case "events occurring at the scale of individual faults" leads to "six intertimes"?

Yes, we agree. So we removed the text referring to the 1st case (events occurring everywhere inside SZ Timpe) since these data are not used in the work or in Part II paper, and maintained only the results regarding the 2nd case (events occurring at scale on individual faults). However, the readers can find further details in Azzaro et al. BGTA 2012 cited in the text.

The inset of Fig. 2, probabilities of occurrence of an earthquake in the next 5 years, refers to the 2nd case (+ why considering a 5 years period?)?

The 5 years period was chosen as representative of short-term earthquake rupture forecast in a high seismic rate region like Etna.

Apparently, faults are assumed to have the same mean recurrence time, but this is not explained.

Now it is clearly specified.

The inset is too small to correctly appreciate the curves.

We fixed this problem by enlarging the inset figure.

The time-dependent model, Brownian Passage Time model, should be introduced.

The text has been now modified and introduced the relevant reference.

The last sentence, referring to a bootstrap analysis, is difficult to understand without more explanations on the test done.

The sentence has been put forward to link it better with the intertimes analysis.

R2. Section 4 Section 4 describes the instrumental earthquake catalog, the delineation of area sources, the determination of seismogenic depths, the estimation of Gutenberg-Richter models for the area source zones, and the estimation of recurrence parameters for a gridded seismicity model. Area sources here are buffer zones around faults. Can you give more precisions on how this buffer zone is delineated (width, association of earthquakes with the fault)?

The text has been slightly modified.

Frequency-magnitude distributions based on the instrumental data is compared to the frequency distribution based on historical data. Why not combining both? E.g. at the scale of the Timpe zone, combining both would lead to a recurrence model fitting both the instrumental magnitudes (interval 2-3.5) and the historical magnitude rates (interval 3.5-5.0), instead of over-estimating slightly the historical rates? Magnitudes larger than 3.5 are contributing strongly in the probabilistic hazard estimation.

The basic assumption we accept in our analysis is the representativeness of few years

of high quality seismic monitoring during an interseismic period for the long-term seismic rates of faults: this assumption on annual/multi-decadal timescale recurrences is, in our opinion, supported by the global agreement of short term occurrences (red dots in Fig. 6), with the ones obtained with the whole catalogue (blue dots). Note that all the rates in Fig. 6 are annual rates, and we added the length of the historical catalogue so that the readers can estimate the completeness of low/high magnitude (we believe the departure of blue dots from the G-R at M∼<3.5 are due to incompleteness); consider that usually this kind of graphs show rates that cannot be linked at all. The inner coherence in treatment suggested us to avoid the simultaneous fitting of both instrumental and historical rates (what to do for Pernicana, for example?).

Nothing is said about the presence of clustered events in the catalog (swarms, foreshocks, aftershocks?). How do the authors handle this issue, which is of importance when establishing earthquake recurrence models and calculating b values?

The extrapolation of a short interseismic period is the key for avoiding declustering in the catalog. Consider that our final aim is modelling a generalized non-Poisson process, as this is more adequate to represent the seismicity in volcanic areas. In terms of seismic moment and number of events, note the regular trend of the cumulative curves in Fig. 3. We modified the figure and slightly the text.

R2. Distributed seismicity Rather than arbitrarily excluding cells where "strange" b-values have been obtained, would it be possible to apply some criteria on the estimation of the b-value, e.g. increase the minimum number of events in the cell or impose a minimum magnitude range available? These criteria would ensure the reliability of the recurrence curve inside each cell. Besides, are the b-values obtained within the values expected for volcanic areas? It is hardy possible to locate the b-values mapped in Fig. 8, without any topography or country border.

Your comment is great, we performed several tests by changing the search radius, minimum number of events, minimum magnitude, and fitting algorithms too. The final

choice here presented is a compromise to have a formal procedure (outliers removal) for selecting automatic results. We added some words on that, and put geographic references on Fig. 8. About the b-values of volcanic areas, as answered to the other referee, we observe the highest ones in the central craters; completeness and detection capability of the network will surely affect the corner toward the sea.

R2. Section 5 The magnitude-size scaling relationship for the Taupo volcanic zone is compared to the relationship for Mt Etna, then both are used for estimating maximum magnitudes. Is it correct to compare 2 relationships established on disjoint datasets (for Taupo, minimum length is larger than 10km, while for Etna maximum length is around 10km) ? Could you add a short discussion on extrapolating scaling relationships?

We used both MSRs for minimizing the epistemic uncertainty associated with them. In any case, the effective interval of extrapolation is narrow, since the length of faults to be used for estimating maximum magnitudes is mostly in the range 7-11 km, i.e. next to the lower part of Taupo MSR (see Fig. 10, frame b, length in logarithm scale). We now added a comment on this aspect.

The section concludes saying that "the mean recurrence times associated to Mmax values vary from 22 to 166 years, periods generally consistent with those historically observed for the individual faults". I don't understand the sentence, as the mean recurrence time estimated from the historical dataset is 71 years (Table 2).

The referee comment is correct. We rephrased the sentence.

How is the aperiodicity factor estimated in this model (Table 2)?

The aperiodicity factor ïĄą, defined as the standard deviation of the recurrence times over their mean, has been estimated by introducing formal error propagation to take into account the uncertainties in Mmax and slip-rates and so to explore how these uncertainties affect the variability in Tmean (details in Peruzza et al., 2010 and Pace et al., 2016).

R2. Other remarks Section 3.1 should be suppressed, as there is no Section 3.2. We put the paragraph (3.1) since we wanted to highlight the topic of the characteristic magnitude and related recurrence time, separating it from the general overview on the historical seismicity.

Section 4, Figure 6: the time period used to estimate annual rates of historical earthquakes for SZ Timpe, FF, STF-SVF, MF-SLF, should be the same? As the completeness of historical data must be homogeneous within the rather small Timpe zone? The time period indicated in the legend, 1832-2015, corresponds to a larger time window, 184 years. Why reducing this time window to 138 or 142 years to calculate annual rates? Please make this point clear, as it is confusing.

Your comment is right, the annual rates for the Timpe area of historical earthquake data are equally setto the large time window 1832-2015, the different years shown in the figure refer to the Tlast-Tfirst event inside each area. For Pernicana fault (PF), we cannot state the same completeness period, as it was a deserted area till late 1970s, and therefore local macroseismic effects are not documented. Again the years indicated (35) refer to the interval between the first and the last event assigned to that area, in the historical earthquake catalogue. We did some effort for better explaining it.

L 248: "This overall picture is consistent with the inter-time distribution of earthquakes (Sicali et a; 2014)" : please, how do you relate b-values with inter-time distributions?

We did not enter into much details with respect to results of the inter-event time (IET) distribution analysis as they can be found in the paper by Sicali et al. (2014). In the text we referred to analogies in the patterns, at different seismogenic depths, of b-values and IET distributions, fixing their seismotectonic significance. The text has been now modified to make clearer this.

Table 2: should be cited in Section 3 dealing with mean recurrence times.

We cited it in Section 3, by renumbering the tables and inverting the columns, as many

data are not comprehensible at this stage we added a disclaimer phrase.

Why providing the "Mmin for which is calculated the probability of occurrence" as it is not mentioned nor discussed in the text?

This is an oversight; Mmin is referred only to the minimum magnitude of the instrumental dataset, and not to the probability of occurrence. We corrected the text in the caption.

L184-185: "This option has a dual purpose: i) to provide a less detailed characterization mediating features inside heterogeneous, " => there must be a word missing?

Yes, the sentence has been rewritten. L 295: "Considering the approximations due to the use of different dimensional measurements, the comparison is fairly explanatory" => exploratory?

The term "explanatory" is correct.

L 321: See tab 1 : should be Table 2?

Yes, we corrected this citation.

R2. Conclusion L350: "Taken as a whole, the FMD of the SZ Timpe is similar to the FMDs and depth distributions of the Moscarello (MF) and S. Leonardo faults (SLF), whilst the Fiandaca Fault (FF), S. Tecla and S; Venerina faults (SVF) show, respectively lower and higher b-values and activity rates." => sentence which is confusing and needs to be re-phrased. MF and SLF belong to the same SZ. Timpe encloses MF-SLF, FF, and STF-SVF, so the seismic rates in Timpe must be higher or equal to the sum of these 3 FMDs.

The sentence is a just comment on the general features of the whole SZ Timpe compared to the individual area sources. However, the difference you observed can be accounted for: i) the SZ Timpe also includes two small triangular areas (see upper right panel in Fig. 6), adding other 183 earthquakes (see also Fig. 4 caption that

reports the number of eqs for each SZ and the whole SZ Timpe); ii) the weight of eqs of MF-SLF in terms of seismic moment released is much higher compared to the ones of FF and STF-SVF, and hence the similarity between SZ Timpe- MF-SLF is more evident. We modified the text accordingly.

Please also note the supplement to this comment:
https://www.nat-hazards-earth-syst-sci-discuss.net/nhess-2017-127/nhess-2017-127-AC1-supplement.zip

---

## Author Comment (AC2) · 28 Jul 2017

Dear Graeme, we appreciated very much your constructive criticism and helpful hints, aimed at improving really the paper; you are acknowledged for your contribution. We answered to all the comments you posed and modified the text accordingly. Hereinafter the detailed list of your comments and our replies, a zip file containing the revised manuscript with tracked changes, and the new figures that have been modified. On behalf of all the authors Raffaele Azzaro

R1. First general consideration: The main topic omitted in this paper is the characterization of epistemic uncertainty in the source model and the manner in which this

is formulated for a PSHA calculation. Whilst some discussion on this topic can be found, albeit briefly, in section 6 of the accompanying Peruzza et al. manuscript, complete omission of the epistemic uncertainties and the combination of the different model approaches presented here diminishes the value of this manuscript as a stand-alone paper on source modelling in volcanic regions. I recommend the authors to consider adding some additional discussion here as to how epistemic uncertainty should be treated and outline the basic formulation of the logic tree. Some overlap with Peruzza et al. (2017) is tolerable in this case.

We thank you for this comment and yes, we agree, some overlapping with Part II paper is useful both for having a stand-alone paper, and for commenting the epistemic uncertainties in source modelling. We entered some new lines in the introduction and a new block of text now marked as Chapter 6, where we also added the picture representing the logic tree approach (taken from Part II paper).

R1. Second general consideration: Secondly, the authors explain in section 2 that destructive historic events have occurred both in periods of activity as well as times of quiescence, and that the recurrence models for slip in larger characteristic events behave in a manner typical of those under tectonic stress rather than local magmatic stress. Whilst it is possible to accept this at face value given the lack of correlation mentioned, it is not so true to say that this applies to all the other seismicity. Within the distributed sources and area zones the recurrence is dependent upon the rate of all seismicity, which will be more closely linked to cycles of eruptive activity and quiescence. In a PSHA risk mitigation context, this means that if considering the probability of exceeding ground motion in a given time period (e.g. 5, 30, 50 years) one needs to account for the probability of an eruptive episode and the probability of exceeding the given levels of ground motion conditional upon the occurrence of the eruptive episode. This is in addition to the baseline hazard during periods of quiescence. Of course, this widens discussion regarding the quantification of the probability of eruptive episodes, but it may be important for putting this work into practice. This is not a critical flaw

in the methodology described, but may be a theoretical limitation of the assumptions made in the application of the recurrence models for the area and distributed seismicity sources.

The reviewer comment is right, in a generalized view of the problem. We added some lines at the end of chapter 2 to widen the discussion and references.

R1. Comments in Detail: Lines 19-20: "We derive a magnitude-size scaling relationship specific for this volcanic area" – Change "specific for" to either "specific to" or "specifically for" Lines 20 – 21: "Pace et al. (2015)" is "Pace et al. (2016)" in bibliography Lines 25: "These analyses to not account regional M>6" – "Do not account for regional . . ." Line 29: "However, apparently less evident . . .", can be changed to just "Less evident but equally . . ." Line 36: comma needed after first "which" Line 37: Should be "computation codes developed for the whole of Italy" Line 62: comma needed after "widespread" and then removed after "eastern flank" Line 82: ". . . seismic hazard applications regards the question of . . ." is better phrased as ". . . seismic hazard applications is the question of . . ." Line 84: "It is a matter of fact that destructive earthquakes in the Timpe area historically occurred both during flank eruptions and not" – Perhaps change the "and not" to "as well as during periods of volcanic quiescence".

Done

Lines 104 -105: "It has to be noted that moderate values of magnitude for heavily damaging events are a feature of seismicity in active volcanic areas such as Etna, whereas in tectonic domains crustal earthquakes producing the same effects are generally associated with M > 6." This comment has particularly significant implications for seismic hazard analysis and I would encourage the authors to: i) add a citation, ii) if known, briefly summarise what are believed to be the potential factors that may explain this observation.

Done

Line 132: Needs comma after "somehow uniform" Line 170: Replace "global" with "European".

Done

Line 180: The use of the detailed areal sources and the extended sources are not clear. Are these alternative branches on an epistemic uncertainty analysis as the comment regarding uncertainty would apply? If so, then the authors need to clarify how the two different models are weighted. If not, then it is unclear how the authors are partitioning the moment rate between the two models.

The extended area source marked in red (SZ Timpe) is given only for a comparison of the parameters with the more detailed SZs; now we clarify this in the text.

R1. Section 4.1.2 The assertion of a Gutenberg Richter model for the various faults is not entirely consistent with the observation shown in Figure 6. In nearly all cases the observed rate of earthquakes around M 3 is greater than that implied by the GR models, which suggests some kind of hybrid characteristic model. This may be shifting the trend toward lower b-values. Did the authors consider a hybrid model in which larger events occur more frequently than predicted by GR? The trend is less obvious for the Timpe zone, which reflects a common perception of GR-behavior across zones spanning larger spatial domains.

The effect commented by the referee is indeed visible only for FF, that is the SZ having the lower number of earthquakes. We do expect also some completeness problems, as far as the seismicity is located at depth and low magnitude events can be missed. We don't consider a hybrid model at this stage of hazard parametrisation, but we will consider it for future implementations.

R1. Section 4.2 The usage of distributed seismicity in this context should be debated more than is done so here. Given the relative brevity of the seismic catalogue, when looking at b-value variation on a fine spatial resolution it may be increasingly likely that

the values in any given cell may reflect a transient process. Even if the variation in b-value is cannot be attributed to statistical artefact, can the authors rule out the possibility that they are related to transient properties of the state of stress around particular elements in the complex volcanic system (including interaction with fluids), even if the period is quiescent? How representative might these values be of recurrence on a multi-decadal timescale?

The reviewer comment is right, it is not possible to calculate the b-value of a cell as a function of time since earthquakes aren't sufficient if split into different time windows. Therefore we can consider only the spatial variation of b-value. This said, we cannot rule out the possibility that transient properties of the state of stress are influencing the b-value. We acknowledge this limitation in the text, by declaring that the basic assumption we accept in this analysis is the representativeness of few years of high quality seismic monitoring during an interseismic period for the long-term seismic rates of faults: this assumption on annual/multi-decadal timescale recurrences is, in our opinion, supported by the global agreement of short term occurrences (red dots in Fig. 6), with the ones obtained with the whole catalogue (blue dots). We did some effort for better explaining this part, a point raised by Reviewer 2 too.

Please also note the supplement to this comment:
https://www.nat-hazards-earth-syst-sci-discuss.net/nhess-2017-127/nhess-2017-127-AC2-supplement.zip

---

## Author Response (AR1)

**Answers to EDITOR**

Dear Oona,

we appreciated very much your comments aimed at improving the final version of the paper; you are acknowledged for your helpful hints. We entirely accepted your suggestions and modified the text accordingly. Hereinafter the detailed list of your comments and our replies, joined to the revised manuscript with tracked changes.

On behalf of all the authors

    Raffaele Azzaro

L.90 A some role of volcanic activity on the occurrence of the low magnitude (M<3) shallow seismicity results from the inter-event times (IET) statistical analysis (Sicali et al., 2014), indicating the central area of the volcano, beneath the summit craters, as the most influenced, seismicity featuring mainly clustered.

--Please re-write

*Done*

L. 147: we calculated intertimes of earthquakes occurring on the same fault (in all, six intertimes).

----In the section above you mention nine events (L 130)? Could you please add a Table xx providing the details of the nine events used to calculate IET? Or at least indicate the events on the map Figure 2?

*We modified Fig. 2 by adding a table listing the events considered in the IET analysis.*

L. 151. boostrap analysis

--you may want to add "bootstrap analysis by sampling with replacement xxx times the initial IET dataset (Table xx)

*Done*

L. 154 Please add to text your answer to the reviewers comment "The 5 years period was chosen as representative of short-term earthquake rupture forecast in a high seismic rate region like Etna".

*Done*

L. 286 We considered only the spatial variation of the b-value as a function of time since the number of earthquakes in the grid nodes is not generally sufficient to be split into different time windows

-- You mean : We considered only the spatial variation of the b-value  since the number of earthquakes in the grid nodes is not generally sufficient to be split into different time windows

*Done*

L.281 Note that in the eastern sector including the SZ Timpe at depths ranging from 2 to 6 km bsl, the b-value pattern varies widely both in number (0.7-1.2) and in space

----It is not possible to situate the SZ Timpe , the crater or the eastern flank on Figure 8. You mention in your reply that you provide a new Figure 8 but maybe you forgot to add it ?? You may want to replace "number" with "value". Please add new Figure 8

*We now modified Fig. 8 by reporting the location of crater and the outlines of SZ Timpe on the geographic grid; we also highlighted the depth value of each section on the block-diagram.*

L. 290. Also the IET analysis (Sicali et al., 2014) exhibits some similarities in the spatial distribution, indicating that features of earthquake occurrence in the western flank-central sector are very different from PF sor SZ Timpe, the latter more related with regional tectonics

Please re-write this sentence, I do not understand it?

*Done*

L. 297 " In conclusion, we cannot rule out the possibility that transient properties of the state of stress may influence the b-value in some 295 cell of our grid but the above comparisons as well as the overall agreement of short term occurrences (red dots in Fig. 6) with the ones obtained by the historical catalogue (blue dots), induced us to accept the assumptions done on the representativeness of few years of instrumental seismicity during an interseismic period, and thus consider the distributed seismicity in the source model."

Suggestion: In conclusion, even if we cannot rule out the possibility that transient properties of the state of stress may influence the b-value in some cells of our grid, we believe that the above comparisons as well as the overall good agreement between short term instrumental and historical catalogue seismicity rates (respectively red and blue dots in Fig. 6), are sufficient evidence that seismicity rates deduced from a few years of instrumental seismicity during an interseismic period are representative of the longer term seismicity rates and can thus be considered to represent the distributed seismicity in the source model.

*Done*

L. 303 approach: it converts

----suggest to replace with "by converting"

*Done*

L. 345 the comparison is fairly explanatory

--- what do you mean?

L. 373 slip-rate values and related uncertainties, from which are strongly dependent (see Fig. 9); Tmean obtained ...☐

---suggestion slip-rate values and related uncertainties,  which are strongly dependent; resulting Tmean..

*Done*

L. 374 However, these values cannot be merely compared with the one obtained from the historical earthquake dataset, which is referred to the SZ Timpe (see Tab Section 3.1)

---- suggestion "..compared with those resulting from the analysis of the historical earthquake dataset (Table xxx), representative of the entire Timpe SZ."

*Done*

L 390 are combined to the areas □

--- replace "to" with "with"

*Done*

L. 393 that are here modelled till a Mmax=4.5.

---Suggestion "where only earthquakes between M=Mmin and Mmax=4.5 are modelled"

*Done*

Typos

L.47 degree of details - change to "detail"

L.217 the reader- change to "reader"

L. 225 - 235 Etna □ - change to Mt Etna

L.257 we calculated alfa - change to α

*Done*

FIGURES

Fig. 2 Major events considered for the analysis are marked in white -- suggestion : are outlined by a white circle. Please provide dates on the map if not in a Table.

*Done; dates of events are indicated in the table annexed to Fig. 2.*

Figure 6: Years indicate the time-window of the historical macroseismic catalogue □

--- Years indicate the actual time-window (Tlast- Tfirst) of the events in each sub-catalogue.

*Done*

Figure 8…you mention a new Figure but the present Figure 8 is the same as the old one.

*Modified according to the suggestion.*

*In addition, we also corrected some typos in Figs. 3, 5 and 9.*

[revised manuscript text omitted]